# Circulating trans fatty acids are associated with prostate cancer in Ghanaian and American men

Tsion Zewdu Minas[1,2,15], Brittany D. Lord [1,15], Amy L. Zhang[1], Julián Candia [1,3], Tiffany H. Dorsey[1], Francine S. Baker [1], Wei Tang [1,4], Maeve Bailey-Whyte[1,5], Cheryl J. Smith[1], Obadi M. Obadi[1], Anuoluwapo Ajao[1], Symone V. Jordan[1], Yao Tettey[6], Richard B. Biritwum[6], Andrew A. Adjei[6], James E. Mensah[6], Robert N. Hoover[7], Ann W. Hsing[8,9], Jia Liu[10], Christopher A. Loffredo[11], Clayton Yates [12,13,14], Michael B. Cook[7] & Stefan Ambs [1] ✉

The association between fatty acids and prostate cancer remains poorly explored in African-descent populations. Here, we analyze 24 circulating fatty acids in 2934 men, including 1431 prostate cancer cases and 1503 population controls from Ghana and the United States, using CLIA-certified mass spectrometry-based assays. We investigate their associations with population groups (Ghanaian, African American, European American men), lifestyle factors, the fatty acid desaturase (*FADS*) genetic locus, and prostate cancer. Blood levels of circulating fatty acids vary significantly between the three population groups, particularly *trans*, omega-3 and omega-6 fatty acids. *FADS1/2* germline genetic variants and lifestyle factors explain some of the variation in fatty acid levels, with the *FADS1/2* locus showing population-specific associations, suggesting differences in their control by germline genetic factors. All *trans* fatty acids, namely elaidic, palmitelaidic, and linoelaidic acids, associated with an increase in the odds of developing prostate cancer, independent of ancestry, geographic location, or potential confounders.

Although prostate cancer has a high global incidence and mortality[1,2], the factors that cause prostate cancer remain incompletely understood. Men of African ancestry, including African American and Ghanaian men, have a disproportionately higher burden of lethal prostate cancer when compared to European American men[1,3]. This increased mortality burden could be partially attributed to unique risk factor profiles in different populations, as well as distinct inflammatory and immune responses that drive cancer aggressiveness and impact survival, as we have shown[4].

[1]Laboratory of Human Carcinogenesis, Center for Cancer Research, National Cancer Institute (NCI), Bethesda, MD, USA. [2]Center for Innovative Drug Development and Therapeutic Trials for Africa, Addis Ababa University, Addis Ababa, Ethiopia. [3]Longitudinal Studies Section, Translational Gerontology Branch, National Institute on Aging, Baltimore, MD, USA. [4]Data Science & Artificial Intelligence, R&D, AstraZeneca, Gaithersburg, MD, USA. [5]School of Medicine, University of Limerick, Limerick, Ireland. [6]University of Ghana Medical School, Accra, Ghana. [7]Division of Cancer Epidemiology & Genetics, NCI, Rockville, MD, USA. [8]Stanford Cancer Institute, Stanford University, Palo Alto, CA, USA. [9]Stanford Prevention Research Center, Stanford University, Palo Alto, CA, USA. [10]Cancer Genomics Research Laboratory, NCI, Rockville, MD, USA. [11]Lombardi Comprehensive Cancer Center, Georgetown University, Washington, DC, USA. [12]Department of Pathology, Johns Hopkins University School of Medicine, Baltimore, MD, USA. [13]Department of Oncology, Sidney Kimmel Comprehensive Cancer Center, Johns Hopkins University School of Medicine, Baltimore, MD, USA. [14]Department of Urology and the James Buchanan Brady Urological Institute, Johns Hopkins University School of Medicine, Baltimore, MD, USA. [15]These authors contributed equally: Tsion Zewdu Minas, Brittany D. Lord. ✉e-mail: ambss@mail.nih.gov

The role of fatty acids in prostate cancer has been studied extensively, but the observations are conflicting, and a consensus of the effects of fatty acids on prostate cancer risk has yet to be achieved[5,6]. A meta-analysis on prospective studies investigating the association of 14 circulating fatty acids, namely saturated and mono- and polyunsaturated fatty acids, with prostate cancer risk in 5098 cases and 6649 controls reported an inverse association between stearic acid, a saturated fatty acid, and the risk to develop prostate cancer. From another meta-analysis, focusing on dietary *trans*-fatty acid, the authors concluded that an elevated total intake of *trans*-fatty acids may increase prostate and colorectal cancer risks[6]. Other studies, using epidemiological and experimental approaches, linked the intake and synthesis of saturated fatty acid to advanced or fatal prostate cancer[7,8]. Together, these studies support the involvement of fatty acids in prostate tumorigenesis and progression. Nevertheless, differences with reference to population groups were not sufficiently explored in these large studies because of a lack of diversity in the assessed populations.

Being aware of this knowledge gap, we decided to characterize the relationship between circulating fatty acids and prostate cancer in the ethnically diverse National Cancer Institute (NCI)-Maryland and NCI-Ghana Prostate Cancer Case-Control studies, with an over-representation of men of African descent. We aimed to find out if common associations exist or if there are distinct patterns among Ghanaian, African American, and European American men. In addition, we explored how circulating fatty acid levels may relate to demographic, lifestyle, and germline genetics, and to an immune-oncology marker profile.

## Results

### Serum fatty acid levels are different between Ghanaian men and men from the United States

It was the aim of our study to gain knowledge whether serum fatty acid profiles and their association with prostate cancer are different between Ghanaian, African American, and European American men. We utilized two case-control studies with an overrepresentation of men of African ancestry: the NCI-Ghana and NCI-Maryland Prostate Cancer Case-Control Studies. Both studies have been previously described[9–11]. Participant characteristics are shown in Supplementary Table 1. A CLIA-certified, mass spectrometry-based assay was applied to measure concentrations of 24 fatty acids (Supplementary Table 2) in sera from 2934 participants, including 1431 prostate cancer cases (585 Ghanaian, 407 African American, 439 European American) and 1503 population controls (658 Ghanaian, 381 African American, 464 European American). To control for any batch effects, the serum samples were assayed in a random order along with 5% blind duplicates. All 24 fatty acids were detected in 100% of the samples tested.

To uncover differences in circulating fatty acid profiles between the three groups of men in our study, we applied unsupervised hierarchical clustering to examine how the levels of the 24 fatty acids may associate with population groups (Fig. 1). We performed this analysis separately for controls (Fig. 1a) and cases (Fig. 1b). Among the controls, fatty acid levels tended to cluster by population group, with marked differences between Ghanaian, African American, and European American participants. A similar pattern was observed among cases, but the separation into Ghanaian men as one group and the African American and European American men as the other group was not as robust. These findings are consistent with the observed differences in median absolute concentrations for the 24 fatty acids between Ghanaian, African American, and European American healthy controls based on group comparisons, with Bonferroni-corrected significance testing to address multiple comparisons (Supplementary Table 3). Additional clustering analyses investigating associations between the 24 fatty acids and Gleason score (dichotomized into ≤ 7, ≥ 8) did not reveal marked clustering patterns related to Gleason score

(Supplementary Fig. 1), suggesting that the clustering of Ghanaian cases away from African American and European American cases is not driven by the higher prevalence of high grade disease among Ghanaians (34%) than African American (17%) or European American men (17%).

In another approach to characterize dissimilarities in circulating fatty acids between these groups of men, we investigated differences in fatty acid levels after grouping them into five structurally distinct classes: saturated, *trans*, *cis*-monounsaturated, omega-3, and omega-6 fatty acids. An initial analysis showed that circulating levels of these fatty acid classes are disparate between Ghanaian, African American, and European American healthy controls (Supplementary Table 4). We then compared one group of men to another (i.e., Ghanaian vs. African American, Ghanaian vs. European American men, or African American vs. European American men) and calculated fatty acid level ratios from these comparisons (Fig. 1c–e). The analysis was performed for both controls and cases and highlights the significantly higher concentrations of circulating omega-3 fatty acids in Ghanaian men, among both controls and cases, when compared to European American and African American men (Fig. 1c, d). A more in-depth statistical evaluation of these comparisons with Bonferroni adjustments can be found in Supplementary Table 5. In contrast to the omega-3 fatty acid observations, *trans* and omega-6 fatty acid levels were consistently higher in European American and African American men, in both controls and cases, when compared to Ghanaian men (Fig. 1c, d). We did not find these stark differences in an analysis that compared European American with African American men (Fig. 1e). However, *trans* and *cis* monounsaturated fatty acid levels tended to be higher in European American compared to African American men among both controls and cases. Lastly, due to the elevated concentration of circulating omega-3 fatty acids in Ghanaian men, together with their rather low serum concentrations of omega-6 fatty acids, these men had the lowest omega 6:3 fatty acid ratio. Notable, the omega 6:3 fatty acid ratio may have implications for prostate cancer progression. A low ratio has been associated with a delay in disease progression[12].

### Association of socio-demographic and clinical characteristics with circulating fatty acids

We investigated the association of various socio-demographic and clinical characteristics that have been reported to be associated with prostate cancer (age, BMI, education, smoking, diabetes, and aspirin use) with serum levels of circulating fatty acids using a multivariable linear regression model, with adjustment for multiple comparisons (Fig. 2, Supplementary Data 1). Aspirin use and education level had few relationships with circulating fatty acids. Levels of saturated fatty acids tended to be negatively associated with age among European American men while omega-6 fatty acids were negatively associated with age in both African American and European American men (Fig. 2). However, there was no relationship with age among the Ghanaian men. In contrast, BMI strongly associated with most of the circulating fatty acids among these men, but less so among men from the United States (U.S.). Lastly, smoking showed positive associations with two omega-6 fatty acids, namely docosapentaenoic-n6 and docosatetraenoic acids, among Ghanaian, African American, and European American men. In an analysis restricted to cases, we explored possible associations between Gleason score and the 24 individual fatty acids using the same multivariable linear regression model but with addition of Gleason score as a covariate, dichotomized as ≤ 7 versus ≥ 8. In this analysis across the three patient groups, Gleason score was not associated significantly with any of the fatty acids analyzed in this study (Supplementary Data 2).

### Association of dietary factors with circulating fatty acids

Next, we examined the relative contribution of diet to the concentration of circulating fatty acids. Previously, we collected nutritional data

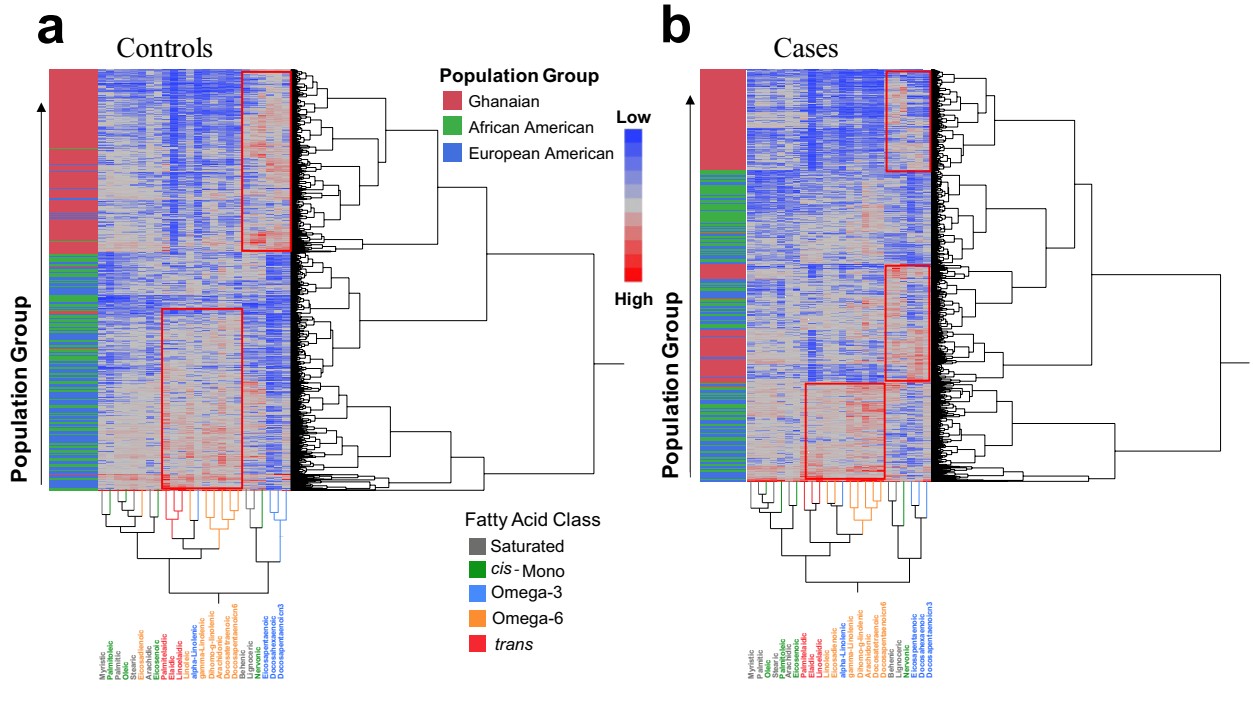

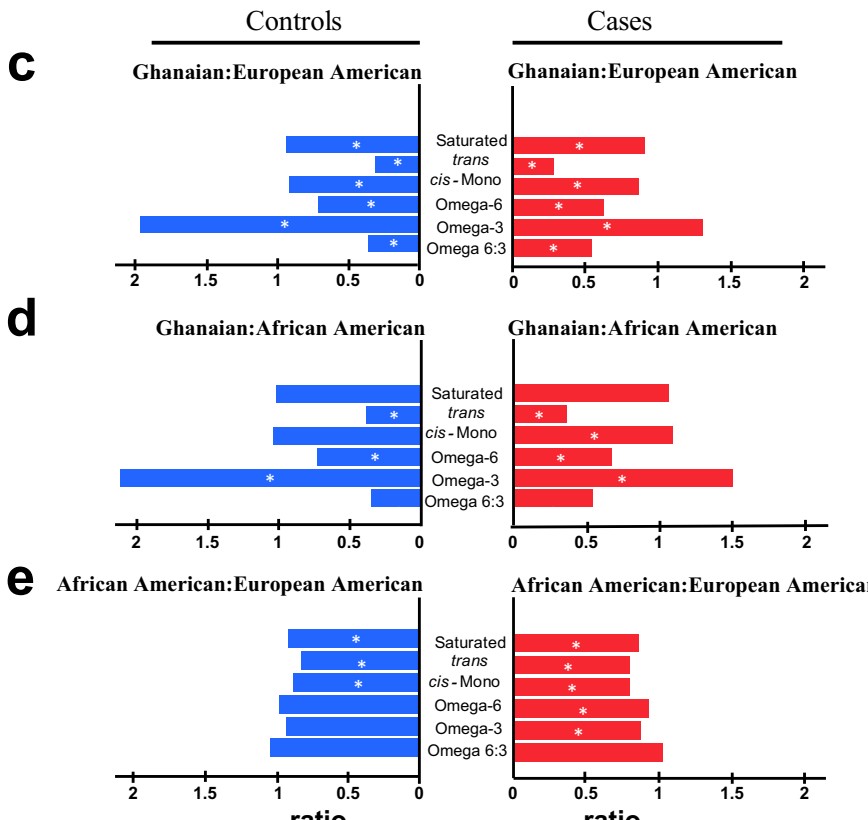

**Nature Communications** | (2023)14:4322

via a brief supplemental questionnaire from participants in the NCI-Maryland study. We applied variance analysis to evaluate the relationship between concentrations of individual fatty acids and diet using the questionnaire data on meat and fat consumption (Supplementary Data 3–5). Few notable associations were observed. In agreement with the literature, fish consumption was significantly associated with the variability of omega-3 and omega-6 fatty acid

levels, accounting for 6.7% and 4.2% of the variability in the levels of docosahexanoic acid in African American cases and controls, and 7.2% and 12.3% of the variability in European American cases and controls, respectively (Supplementary Fig. 2). Frequent intake of bacon fat or drippings during the 2-year period prior to interview significantly, albeit modestly, explained the variability in the concentration of all three *trans* fatty acids only among cases: palmitelaidic (African

**Fig. 1 | Circulating fatty acid levels by fatty acid class and population group in the NCI-Maryland and NCI-Ghana cohorts.** Heatmaps depicting unsupervised hierarchical clustering using absolute (mean, μg/mL) concentrations from individual fatty acids in (**a**) controls ($n = 1503$) and (**b**) cases ($n = 1431$) with categorization into three population groups. Green indicates lower concentrations and red indicates higher concentrations for each participant. Samples were labeled by population group and includes Ghanaian (red, $n = 1243$), African American (green, $n = 788$), and European American (blue, $n = 903$) men. Individual fatty acids were color-coded by fatty acid class (Saturated, grey; *cis*-Monounsaturated, green; Omega-3, blue; Omega-6, orange; and *trans*, red). Red boxes indicate areas of fatty acid concentrations that significantly differ by population group; **c**–**e** Show ratios of mean concentrations by fatty acid class for controls (blue) and cases (red) comparing (**c**) Ghanaian vs. European American, (**d**) Ghanaian vs. African American, and (**e**) African American vs. European American men. Two-sided student's t-tests were used for significance testing of differences in fatty acid levels (by class) between population groups (see Supplemental Table 5 for Bonferroni adjustment). An asterisk (*) indicates a statistical significant difference between population groups after adjustment for multiple comparison. Source data are provided as a Source Data file.

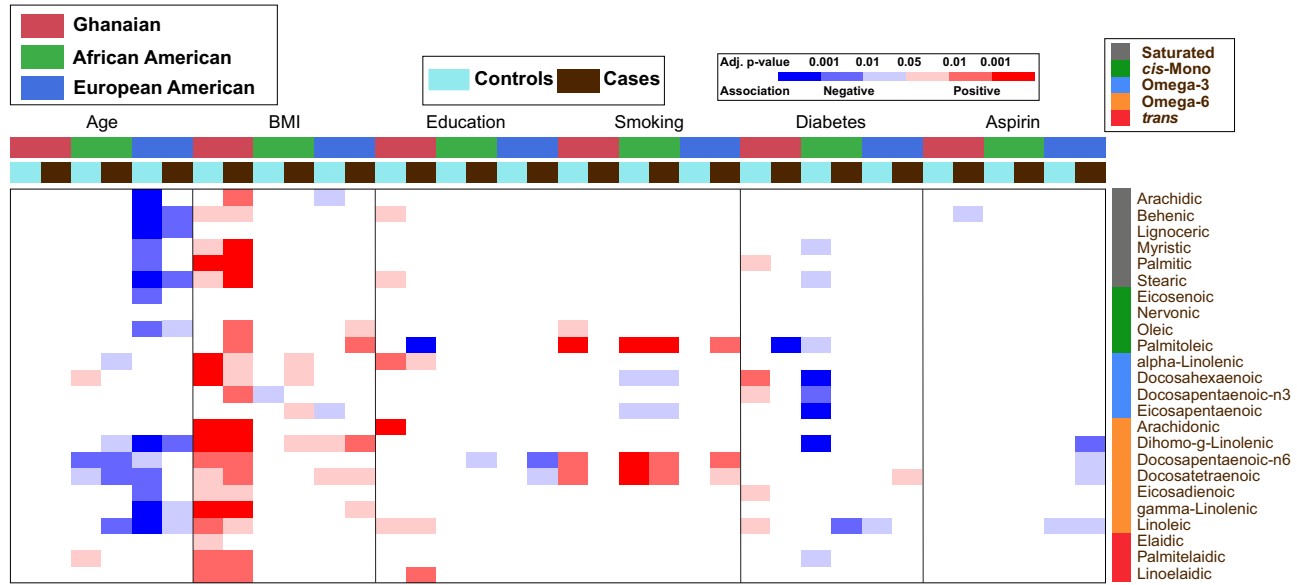

**Fig. 2 | Association of socio-demographic and clinical characteristics with circulating fatty acids in Ghanaian, African American, and European American men with and without prostate cancer.** The association of the 24 fatty acids (as continuous variables) with age, BMI, education, smoking, diabetes, and aspirin use was assessed in prostate cancer cases in brown [Ghanaian (red, $n = 585$), African American (green, $n = 407$), and European American (blue, $n = 439$)] and controls in light blue [Ghanaian (red, $n = 658$), African American (green, $n = 381$), and European American (blue, $n = 464$)] using a multivariable linear regression test. $P$ values were adjusted for multiple comparisons. An analyte was considered significantly associated with socio-demographic and clinical covariables if the multivariable model yielded a false discovery rate (FDR)-adjusted $P < 0.05$ on the F-statistic. Blue represents a statistically significant negative association while red represents a statistically significant positive association. White represents an association that was not statistically significant. The significance level (FDR-adjusted two-sided $P$ value-based) for each association is color-coded. Fatty acids are color coded to indicate their fatty acid class (Saturated, grey; *cis*-Monounsaturated, green; Omega-3, blue; Omega-6, orange; and *trans*, red). BMI = Body Mass Index. Source data are provided as a Source Data file.

American cases: 4.5%, European American cases: 5.0%), elaidic (African American cases: 2.9 %, European American cases: 3.7%), and linoelaidic (African American cases: 2.8%, European American cases: 3.9%) (Supplementary Fig. 3). Thus, dietary differences appear to account for at least some of the variability in the concentration of circulating fatty acids in our cohorts. However, the detected effect sizes were small.

### Association of fatty acid desaturase 1 and 2 (*FADS1/2*) locus with circulating fatty acid levels

It has been shown that the levels of circulating fatty acids are partly under genetic regulation[13,14]. Thus, we examined how germline genetic factors may relate to fatty acid concentrations in our diverse cohorts focusing on key examples from the published literature. We concentrated our efforts on single nucleotide polymorphisms in the *FADS1/2* locus that have been found to influence fatty acid levels in humans[13,14]. In individuals of European descent, this gene cluster has been shown to have the strongest effect among all genetic loci on the levels of certain fatty acids, namely omega-3 and omega-6 polyunsaturated fatty acids, as defined by their enzymatic activity. We selected three single nucleotide polymorphisms (SNPs) covering the *FADS1/2* locus (Supplementary Table 6). We then tested if the levels of each of the 24 fatty acids in blood circulation are influenced by the selected SNPs in our cohort, expecting that only certain

polyunsaturated fatty acids would show associations. We found that the SNPs had significant associations with the levels of several omega-6 fatty acids including arachidonic, dihomo-gamma-linolenic, docosatetraenoic (or adrenic), and γ-linolenic acids in European American men, consistent with the literature, explaining up to 19% of their variance in this population. However, with the exception of rs174556 SNP in *FADS1* gene, which explained a small fraction of the variability (4.7%) in amounts of circulating arachidonic acid among African American cases, these SNPs did not influence the levels of omega-6 fatty acids in African American or Ghanaian men (Fig. 3, Supplementary Fig. 4, Supplementary Tables 7–12). The observation suggests population differences in the genetic regulation of circulating fatty acids involving the *FADS1/2* locus.

### Circulating fatty acid levels and odds of prostate cancer across population groups

We next assessed the association of individual fatty acids and fatty acid classes with prostate cancer in all men, and then stratified by population group (Table 1, Supplementary Tables 13, 14). Notably, three of the four omega-3 fatty acids, namely docosahexaenoic (DHA), docosapentaenoic-n3 (DPA), and eicosapentaenoic (EPA) acids, were inversely associated with prostate cancer among Ghanaian men (Table 1, Fig. 4a). On the other hand, several omega-6 fatty acids were

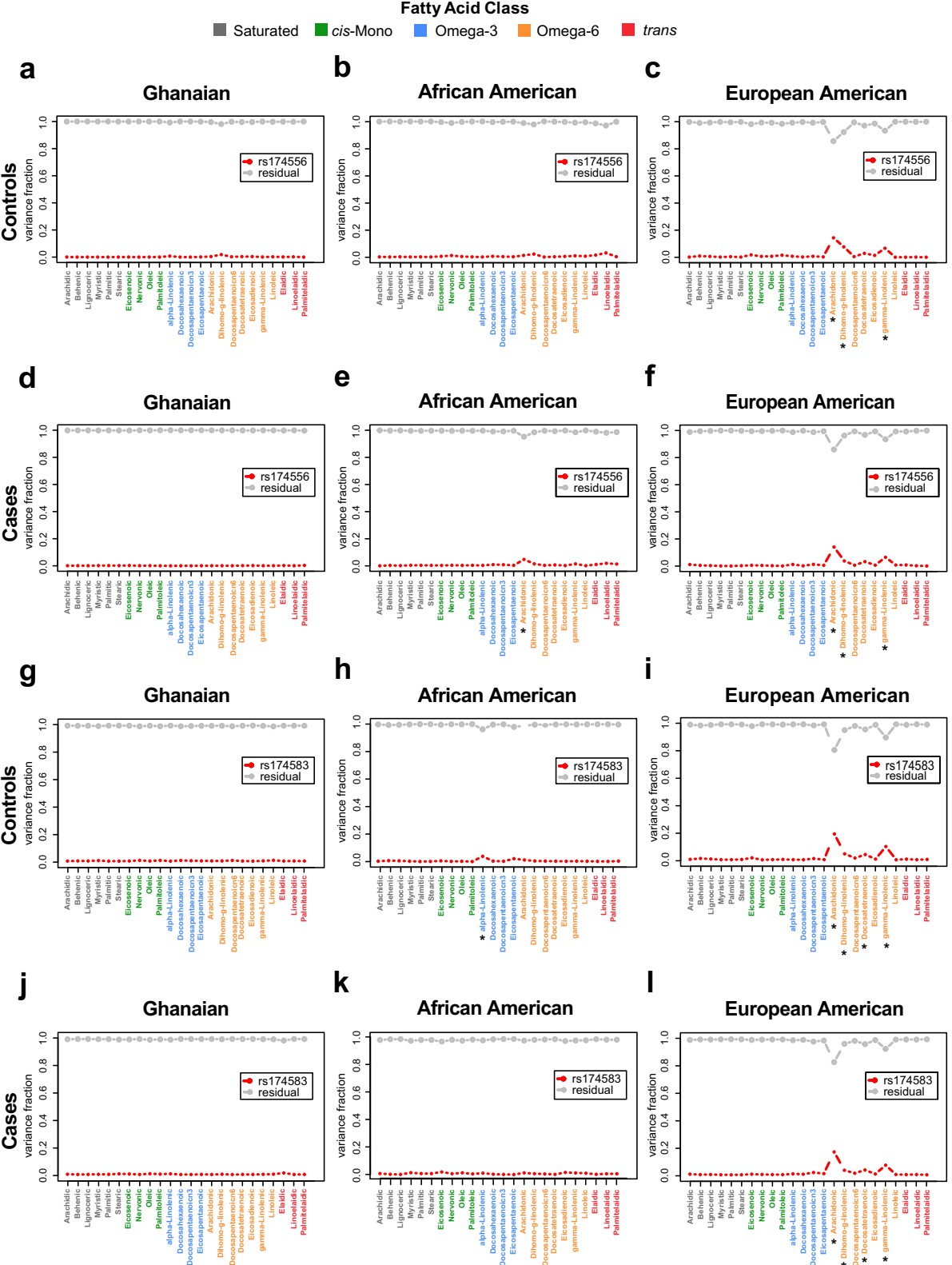

**Fig. 3 | Influence of single nucleotide polymorphisms (SNPs) in the *FADS1* and *FADS2* genes on circulating levels of fatty acids.** Variance analysis for the levels of each of the 24 fatty acids are assessed as a function of a SNP (rs174556) in *FADS1* (**a**–**f**) and a SNP (rs174583) in *FADS2* (**g**–**l**) among (**a**, **g**) Ghanaian controls (*n* = 602), (**d**, **j**) Ghanaian cases (*n* = 511), (**b**, **h**) African American controls (*n* = 350), (**e**, **k**) African American cases (*n* = 344), **c**, **i** European American controls (*n* = 394), and (**f**, **l**) European American cases (*n* = 362). Red dotted line represents the proportion of variance in the levels of the fatty acids that can be explained by the SNPs while the grey line represents the residual variance that remains to be explained by other factors. Asterisks (*) indicate variances in fatty acid levels that are significantly explained by the SNP using an Analysis of Variance statistical test. Bonferroni-adjusted significance threshold of *P* = 0.002 is used to account for multiple testing. Fatty acids are color coded to indicate their fatty acid class (Saturated, grey; *cis*-Monounsaturated, green; Omega-3, blue; Omega-6, orange; and *trans*, red). Source data are provided as a Source Data file.

**Table 1 | Association of serum levels of 24 fatty acids with odds of prostate cancer case status stratified by population group**

| Fatty acid | Fatty acid class | All (n = 2934) | | Ghanaian (n = 1243) | | African American (n = 788) | | European American (n = 903) | |
|---|---|---|---|---|---|---|---|---|---|
| | | Multivariable[a] OR (95% CI) | P-value[b] | Multivariable[c] OR (95% CI) | P-value[b] | Multivariable[c] OR (95% CI) | P-value[b] | Multivariable[c] OR (95% CI) | P-value[b] |
| Eicosenoic | cis-Monounsaturated | 1.00 (0.93, 1.08) | 0.930 | 1.08 (0.89, 1.30) | 0.422 | 0.82 (0.71, 0.96) | 0.015 | 1.07 (0.96, 1.20) | 0.235 |
| Nervonic | cis-Monounsaturated | 1.18 (1.09, 1.28) | 1.70E-05 | 1.19 (1.06, 1.34) | 0.004 | 1.18 (0.98, 1.43) | 0.087 | 1.09 (0.91, 1.30) | 0.338 |
| Oleic | cis-Monounsaturated | 1.17 (1.08, 126) | 8.70E-04 | 1.36 (1.15, 1.61) | 3.40E-03 | 0.97 (0.84, 1.11) | 0.626 | 1.17 (1.04, 1.33) | 0.012 |
| Palmitoleic | cis-Monounsaturated | 1.07 (0.99, 1.15) | 0.087 | 0.87 (0.74, 1.02) | 0.092 | 1.07 (0.92, 1.24) | 0.384 | 1.24 (1.07, 1.43) | 0.004 |
| alpha-Linolenic (ALA) | Omega-3 | 1.09 (0.99, 1.19) | 0.074 | 1.75 (1.05, 2.90) | 0.031 | 0.92 (0.80, 1.06) | 0.247 | 1.10 (0.96, 1.25) | 0.179 |
| Docosahexaenoic (DHA) | Omega-3 | 0.51 (0.45, 0.56) | 1.40E-33 | 0.37 (0.31, 0.44) | 4.60E-25 | 0.84 (0.59, 1.19) | 0.323 | 1.01 (0.77, 1.30) | 0.969 |
| Docosapentaenoic - n3 (DPA) | Omega-3 | 0.73 (0.67, 0.79) | 5.00E-12 | 0.53 (0.46, 0.61) | 2.30E-15 | 0.93 (0.77, 1.11) | 0.412 | 1.16 (0.99, 1.36) | 0.066 |
| Eicosapentaenoic (EPA) | Omega-3 | 0.62 (0.56, 0.68) | 1.60E-17 | 0.56 (0.48, 0.64) | 9.30E-11 | 0.68 (0.48, 0.96) | 0.031 | 0.95 (0.77, 1.18) | 0.662 |
| Arachidonic (AA) | Omega-6 | 0.98 (0.89, 1.07) | 0.586 | 0.56 (0.46, 0.70) | 6.00E-06 | 1.03 (0.87, 1.21) | 0.771 | 1.11 (0.95, 1.31) | 0.186 |
| Dihomo-g-linolenic (DGLA) | Omega-6 | 1.18 (1.09, 1.28) | 1.20E-04 | 0.88 (0.75, 1.03) | 0.117 | 1.15 (0.99, 1.34) | 0.064 | 1.30 (1.13, 1.50) | 2.30E-04 |
| Docosapentaenoic - n6 | Omega-6 | 1.09 (1.01, 1.18) | 0.025 | 0.57 (0.46, 0.71) | 6.80E-05 | 1.19 (1.04, 1.35) | 0.009 | 1.25 (1.08, 1.46) | 0.004 |
| Docosatetraenoic | Omega-6 | 1.25 (1.14, 1.37) | 1.30E-04 | 0.84 (0.63, 1.11) | 0.210 | 1.14 (0.99, 1.32) | 0.066 | 1.33 (1.13, 1.56) | 0.001 |
| Eicosadienoic | Omega-6 | 1.09 (1.00, 1.18) | 0.054 | 0.81 (0.65, 1.00) | 0.052 | 0.98 (0.87, 1.11) | 0.742 | 1.27 (1.09, 1.47) | 0.002 |
| gamma-Linolenic (GLA) | Omega-6 | 1.13 (1.04, 1.23) | 0.005 | 0.80 (0.65, 0.97) | 0.027 | 1.07 (0.92, 1.24) | 0.375 | 1.22 (1.07, 1.39) | 0.003 |
| Linoleic (LA) | Omega-6 | 1.03 (0.94, 1.12) | 0.533 | 0.77 (0.63, 0.93) | 0.009 | 0.88 (0.75, 1.03) | 0.103 | 1.12 (0.98, 1.29) | 0.085 |
| Arachidic | Saturated | 1.09 (1.01, 1.18) | 0.033 | 1.35 (1.11, 1.64) | 0.002 | 0.80 (0.67, 0.95) | 0.013 | 1.08 (0.96, 1.23) | 0.209 |
| Behenic | Saturated | 1.08 (1.01, 1.17) | 0.033 | 1.22 (1.06, 1.41) | 0.006 | 0.88 (0.77, 1.01) | 0.078 | 1.04 (0.91, 1.18) | 0.564 |
| Lignoceric | Saturated | 1.02 (0.95, 1.10) | 0.596 | 1.05 (0.92, 1.19) | 0.444 | 0.93 (0.80, 1.08) | 0.326 | 0.99 (0.86, 1.14) | 0.865 |
| Myristic | Saturated | 0.98 (0.91, 1.06) | 0.675 | 0.80 (0.66, 0.97) | 0.021 | 0.96 (0.84, 1.10) | 0.559 | 1.17 (1.03, 1.34) | 0.018 |
| Palmitic | Saturated | 1.22 (1.13, 1.33) | 3.50E-05 | 1.44 (1.22, 1.70) | 9.90E-04 | 1.02 (0.89, 1.18) | 0.739 | 1.20 (1.05, 1.37) | 0.006 |
| Stearic | Saturated | 1.30 (1.20, 1.41) | 7.10E-08 | 1.59 (1.34, 1.89) | 5.90E-06 | 1.04 (0.90, 1.19) | 0.599 | 1.21 (1.06, 1.38) | 0.004 |
| Elaidic | trans-Monounsaturated | 1.52 (1.36, 1.70) | 1.38E-04 | 12.29 (4.77, 31.60) | 1.38E-04 | 1.41 (1.19, 1.68) | 1.73E-04 | 1.39 (1.21, 1.60) | 2.92E-04 |
| Palmitelaidic | trans-Monounsaturated | 1.35 (1.23, 1.47) | 7.00E-10 | 1.69 (1.36, 2.10) | 1.20E-04 | 1.31 (1.11, 1.55) | 0.001 | 1.24 (1.10, 1.40) | 3.50E-04 |
| Linoelaidic | trans-Polyunsaturated | 1.49 (1.35, 1.66) | 1.19E-04 | 2.79 (1.91, 4.07) | 1.19E-04 | 1.31 (1.10, 1.56) | 0.003 | 1.34 (1.18, 1.53) | 4.90E-06 |

[a]Logistic regression adjusted for age at recruitment and population group. ORs represent an increase in estimated prostate cancer risk per one standard deviation increase for each fatty acid.

[b]Bonferroni adjusted threshold for significance is P = 0.0021 to account for multiple testing.

[c]Logistic regression adjusted for age at recruitment. ORs represent an increase in the estimated odds of prostate cancer per one standard deviation increase for each fatty acid.

OR Odds Ratio, CI Confidence Interval.

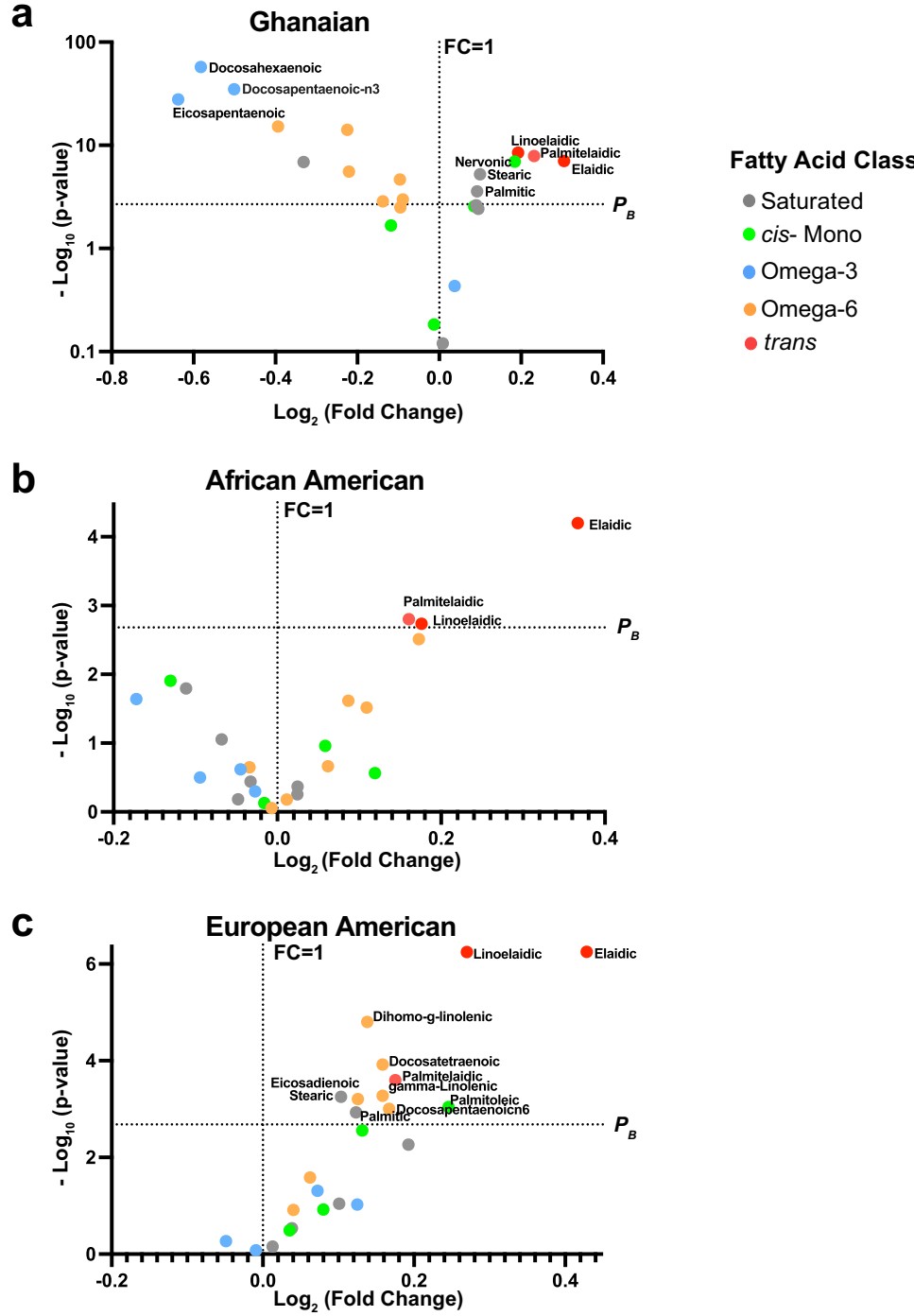

**Fig. 4 | Volcano plot highlighting circulating fatty acids with the most significant differences in their serum levels comparing cases with controls.** Shown are Volcano plots defined by fold difference between cases and controls (log2 [fold change]) (x-axis) and the *P* value from a two-sided Student's t-test (-log10 [*p*-value]) (y-axis) assessing differences for (**a**) Ghanaian (*n* = 1243), (**b**) African American (*n* = 788), and (**c**) European American (*n* = 903) men. The horizontal dashed line represents a Bonferroni-adjusted *P*-value (*P_B*) of 0.002 with differences above this line being significant after adjustment for multiple comparisons. The vertical dashed line represents a fold change of 1, meaning the fatty acid levels are equal in cases and controls. Fatty acids found to the right of the vertical dashed line are elevated in the cases whereas those to the left are lower among them when compared to the controls. Fatty acids are color coded to indicate their fatty acid class (Saturated, grey; *cis*-Monounsaturated, green; Omega-3, blue; Omega-6, orange; and *trans*, red). FC= Fold Change. Source data are provided as a Source Data file.

positively associated with prostate cancer among European American men (Table 1, Fig. 4c). However, only the three measured *trans* fatty acids, namely elaidic, palmitelaidic, and linoelaidic acids, were positively associated with the odds of having prostate cancer in all men combined and across the three population groups (Fig. 4a–c).

The link between *trans* fatty acids and prostate cancer was further investigated using multivariable logistic regression analyses with adjustments for potential confounders. Here, we divided each *trans* fatty acid concentration into tertiles, termed low, intermediate, and high, and assessed the associations with prostate cancer across the three population groups (Table 2). We found a significant dose-dependent increase in the odds of having prostate cancer with increasing elaidic, palmitelaidic, and linoelaidic fatty acid concentrations in all three population groups. Notably, although Ghanaian men

**Table 2 | Association of *trans* fatty acids (elaidic acid, palmitelaidic acid, and linoelaidic acid) with odds of prostate cancer case status**

| | Ghanaian | | | | African American | | | | European American | | | |
|---|---|---|---|---|---|---|---|---|---|---|---|---|
| | median concentration (µg/ml) | Control N | Case N | Multivariable[a] OR (95% CI) | median concentration (µg/ml) | Control N | Case N | Multivariable[a] OR (95% CI) | median concentration (µg/ml) | Control N | Case N | Multivariable[a] OR (95% CI) |
| **Elaidic acid** | | | | | | | | | | | | |
| Low | 2.89 | 219 | 143 | Ref. | 9.93 | 154 | 99 | Ref. | 10.46 | 128 | 70 | Ref. |
| Intermediate | 4.11 | 219 | 169 | 1.55 (1.03, 2.35) | 17.06 | 122 | 128 | 1.83 (1.25, 2.68) | 17.67 | 160 | 134 | 1.38 (0.94, 2.03) |
| High | 6.54 | 220 | 273 | 1.92 (1.30, 2.84) | 34.41 | 105 | 180 | 2.98 (2.04, 4.34) | 33.25 | 176 | 235 | 2.17 (1.51, 3.12) |
| $P_{trend}$ | | | | 0.001 | | | | 1.80E-08 | | | | 1.20E-05 |
| **Palmitelaidic acid** | | | | | | | | | | | | |
| Low | 1.73 | 220 | 105 | Ref. | 2.86 | 172 | 140 | Ref. | 3.08 | 109 | 75 | Ref. |
| Intermediate | 2.71 | 218 | 178 | 1.51 (0.98, 2.32) | 4.44 | 130 | 135 | 1.43 (1.00, 2.05) | 4.57 | 153 | 139 | 1.31 (0.89, 1.93) |
| High | 4.25 | 220 | 302 | 2.26 (1.50, 3.39) | 6.76 | 79 | 132 | 2.55 (1.73, 3.77) | 7.15 | 202 | 22 | 1.49 (1.04, 2.14) |
| $P_{trend}$ | | | | 6.90E-05 | | | | 3.10E-06 | | | | 0.035 |
| **Linoelaidic acid** | | | | | | | | | | | | |
| Low | 3.50 | 219 | 153 | Ref. | 5.91 | 178 | 125 | Ref. | 6.11 | 104 | 61 | Ref. |
| Intermediate | 4.72 | 220 | 148 | 0.97 (0.64, 1.46) | 8.80 | 117 | 158 | 2.17 (1.52, 3.10) | 9.06 | 165 | 129 | 1.26 (0.84, 1.88) |
| High | 6.70 | 219 | 284 | 1.52 (1.03, 2.23) | 14.01 | 86 | 124 | 2.34 (1.59, 3.45) | 14.81 | 195 | 249 | 1.93 (1.32, 2.82) |
| $P_{trend}$ | | | | 0.025 | | | | 7.10E-06 | | | | 1.70E-04 |

[a]Logistic regression adjusted for age at recruitment, BMI, education, diabetes, smoking history, and aspirin use Elaidic acid tertile cutoffs for NCI-Maryland cohort were 13.51 and 22.69 µg/ml.
Elaidic acid tertile cutoffs for NCI-Ghana cohort were 3.47 and 4.98 µg/ml Palmitelaidic acid tertile cutoffs for NCI-Maryland cohort were 3.70 and 5.51 µg/ml Palmitelaidic acid tertile cutoffs for NCI-Ghana cohort were 2.21 and 3.25 µg/ml Linoelaidic acid tertile cutoffs for NCI-Maryland cohort were 7.37 and 10.84 µg/ml Linoelaidic acid tertile cutoffs for NCI-Ghana cohort were 4.09 and 5.34 µg/ml.
Tertile cutoff were determined using distribution of the fatty acids in the control population for each study population, OR Odds Ratio, CI Confidence Interval.
$P_{trend} \leq 0.05$ indicate significant associations in the multivariable logistic regression analysis.

were found to have the lowest mean *trans* fatty acid concentration when compared to African American and European American men (Supplementary Fig. 5), they still experienced significantly elevated odds of developing prostate cancer with increasing *trans* fatty acid levels.

To assess if fatty acid levels were associated with disease severity, we correlated individual fatty acids and fatty acid classes with National Comprehensive Cancer Network (NCCN) risk scores that describe disease severity[15], which were obtained for the African American and European American patients in the NCI-Maryland study[4]. In this analysis, only palmitoleic acid showed a positive dose-dependent relationship with increasing NCCN risk scores, even after adjusting for multiple testing ($P_{\text{trend}}$ = 0.002, Supplementary Table 15).

### Associations of circulating fatty acids with an immune-oncology marker profile

Circulating fatty acids may influence the immune environment. For the NCI-Maryland and NCI-Ghana studies, 82 immune-oncological markers have previously been measured and grouped into pathways: apoptosis/cell killing, autophagy/metabolism, chemotaxis/trafficking to tumor, suppression of tumor immunity (Th2 response, tolerogenic), promotion of tumor immunity (Th1 responses), vasculature[4]. Thus, we assessed whether there was a relationship between the immune-oncology marker-defined pathways and circulating fatty acid levels. For this analysis, we grouped the fatty acids into the five classes as already described and then assessed the relationships between fatty acid classes and pathway activity scores (see methods) by population group and for men with and without prostate cancer. As shown in Fig. 5a, the correlation heatmaps covering the controls revealed significant negative associations between the omega-6 fatty acid class and several of the immune-oncological pathways, as well as a positive association between monounsaturated fatty acids and the vasculature pathway, but only among Ghanaian men (Fig. 5a, Bonferroni-corrected $P < 0.01$). In Ghanaian men with prostate cancer, there were significant positive associations between the omega-3 and omega-6 fatty acid classes and the immune-oncological pathways, and significant negative associations between the omega 6:3 ratio and the activity scores for these pathways (Fig. 5b, Bonferroni-corrected $P < 0.01$). For European American men with prostate cancer, we observed significant positive associations between monounsaturated fatty acids and the apoptosis, inflammation, suppression of tumor immunity and vasculature pathways. In addition, saturated fatty acids associated similarly with the pathway activity score for suppression of tumor immunity and the omega 6:3 ratio with the score for promotion of tumor immunity (Fig. 5b, Bonferroni-corrected $P < 0.01$ for all).

### Discussion

Our study comprehensively characterized circulating fatty acid levels across three distinct population groups, Ghanaian, African American, and European American men, and assessed their associations with prostate cancer. Importantly, circulating fatty acid levels are surrogates for their prostate tissue concentrations, as shown recently[16]. Here, we report that circulating fatty acid levels are different between Ghanaian men and African American and European American men from the U.S., particularly *trans*, omega-3 and omega-6 fatty acids. This may relate to differences in diet, with a lower intake of *trans* fatty acids among Ghanaian men since they eat fewer processed foods. As a main finding, our study suggests a significant positive association of *trans* fatty acid intake with prostate cancer in these three groups of men. Another key observation suggests formerly unrecognized population differences in the genetic control of circulating fatty acids. Multiple studies have shown that common genetic variants in the *FADS1/2* gene cluster exert a significant effect on the circulating levels of poly-unsaturated fatty acids in European descent populations[13,14]. While we replicated these findings for omega-6 fatty acids among European

American men in our study, we did not find the same variants to be associated with omega-6 fatty acids in either African American or Ghanaian men.

Serum fatty acid levels were different between Ghanaian men and African American and European American men from the U.S., with omega-3 fatty acids being consistently elevated in Ghanaian men, in both controls and cases, whereas *trans* and omega-6 fatty acid levels were consistently higher in European American and African American, in both controls and cases. We did not find these differences comparing African American with European American men, suggesting that most of the fatty acid-related disparities between Ghanaian men and men from the U.S. are not due to ancestral genetic factors, but rather due to food intake differences. The omega-3 fatty acids that were elevated in Ghanaian men are typically plant- or marine-derived[17,18]. They are important ingredients of a healthy, traditional diet and are thought to have anti-inflammatory effects that provide cardiovascular benefits[17,18]. Their potentially beneficial role in cancer development remains uncertain and an association with prostate cancer has not been firmly established[5]. We are not aware that others have described differences in circulating fatty acid levels between Ghanaian men and men from the U.S. However, one study reported plasma fatty acid levels for 48 African American cases and 96 controls and 66 Nigerian cases and 226 controls[19]. While not as comprehensive in design as our study, the authors also reported that *trans* and ω-6 fatty acid levels were higher in African American men than in the indigenous African men, here Nigerian men. Additionally, consistent with our findings, omega-3 fatty acid levels were elevated in Nigerian men.

We observed that circulating elevated *trans* fatty acid levels were associated with higher odds of prostate cancer in the three distinct population groups of Ghanaian, African American, and European American men, identifying *trans* fatty acids as potential prostate cancer risk factors independent of ancestry or geographic location. This observation agrees with the findings from a recent meta-analysis linking elevated *trans* fatty acids to an increased risk of prostate cancer[6]. Previous studies in experimental systems have shown that *trans* fatty acid may have pro-tumorigenic properties[20–23], providing plausibility for observations from cancer epidemiology linking *trans* fatty acids to several human cancers[6]. *Trans* fatty acids are unsaturated fatty acids with at least one double bond in the *trans* configuration, and are commonly found in fast foods, highly processed snacks, baked goods, and hydrogenated oils such as margarine[24,25]. As these foods are common in the Western diet, high consumption of *trans* fatty acids have long been studied as a risk factor for cancer[26–28] and adverse cardiovascular events[29–31]. Our study identified significant and consistent associations with prostate cancer for all *trans* fatty acids (16:1, 18:1, 18:2) tested across our three population groups. Multiple investigators have studied the association of these *trans* fatty acids with prostate cancer and most, but not all, reported positive associations[32–34]. A study from the β-Carotene and Retinol Efficacy Trial (CARET), examining the association between individual *trans* fatty acids and prostate cancer risk, showed that C:18 *trans* fatty acids, such as elaidic and linoelaidic acids, were associated with an increased prostate cancer risk, but they did not see this association with C:16 *trans* fatty acids[32].

Circulating *trans*-fatty acids can have various sources. In an Asian Indian population, it was shown that heating and frying with edible oils (including refined soybean oil, groundnut oil, olive oil, and rapeseed oil) resulted in a significant increase in the amount of both *trans* and saturated fatty acids from these oils[35]. In addition, a cross-sectional study in Uganda revealed high proportions of saturated, cis-mono-unsaturated, and *trans* fatty acids, likely due to the high consumption of palm oil and hydrogenated fats[36]. In the U.S. and many other countries, there are two main types of *trans* fatty acids: industrial and natural (ruminant) *trans* fatty acids. Industrial *trans* fatty acids, such as

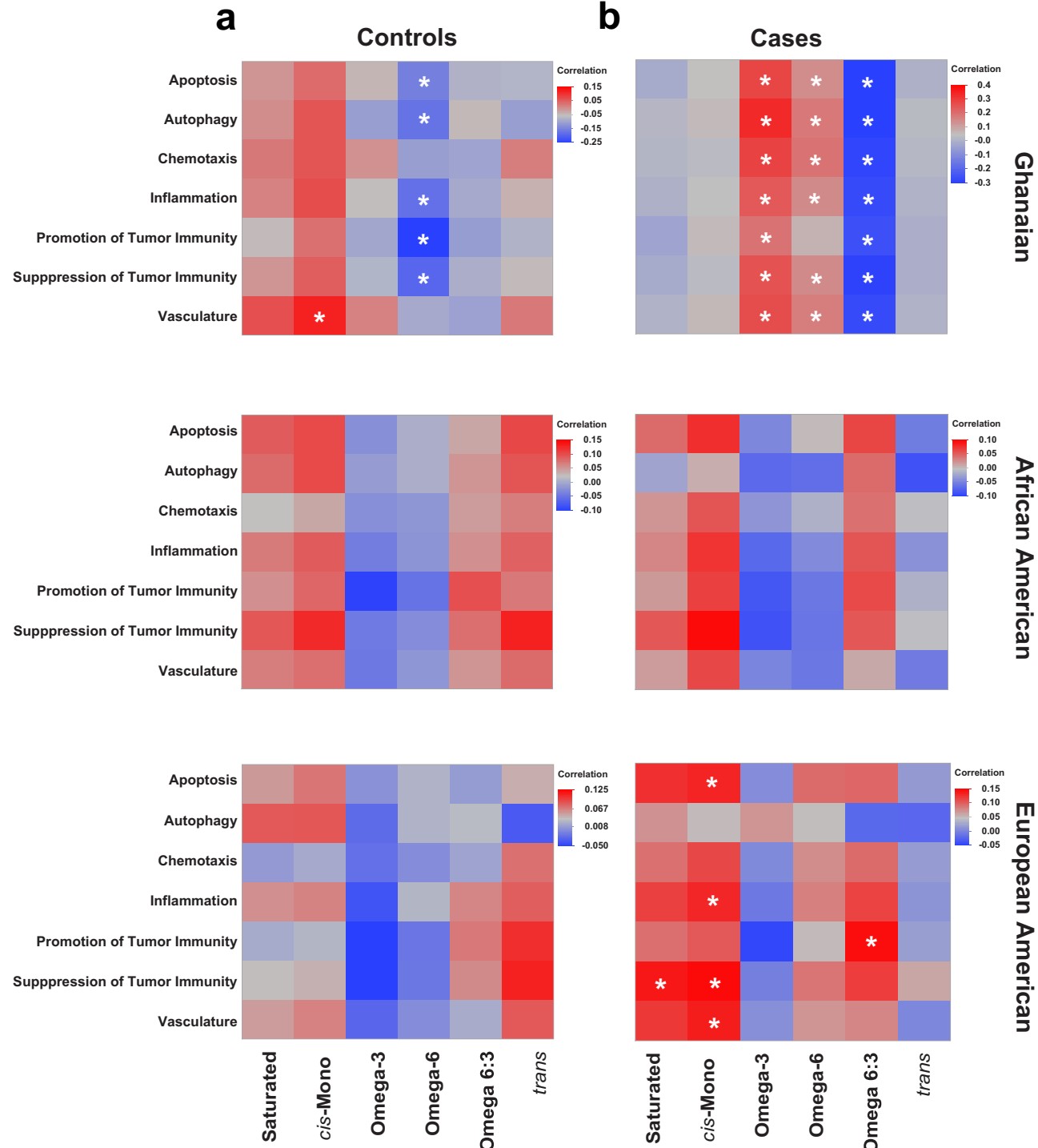

**Fig. 5 | Correlations between serum proteome-defined pathway activity scores and levels of circulating fatty acids grouped by class.** Heatmaps detailing the correlations between seven immune-oncological pathways and seven fatty acid classes in (**a**) Ghanaian (*n* = 653), African American (*n* = 373), European American (*n* = 454) controls and (**b**) Ghanaian (*n* = 488), African American (*n* = 394), European American (*n* = 425) prostate cancer cases. Pairwise linear correlation tests were used to investigate the associations, with asterisks (*) indicating significant relationships using a Bonferroni corrected *P*-value of 0.007 as threshold to account for multiple comparisons. Red = positive correlation, green = negative correlation. For the definition of pathway activity scores, see Methods. Source data are provided as a Source Data file.

elaidic acid, are found in processed foods that contain partially hydrogenated vegetable oils, which are used in frying oils, margarines, spreads, and in bakery products[37]. Because the level of elaidic acid in the blood correlates with intake of highly processed foods, elaidic acid is considered a marker of processed food intake[38]. On the other hand, natural (ruminant) *trans* fatty acids, such as palmitelaidic acid, are made by bacterial metabolism of polyunsaturated fatty acids in the rumen of ruminants such as cows, sheep, and goats, and are present in fats sourced from these animals[37]. In our study, frequent intake of bacon fat or drippings partly explained the variability in the concentration of both types of *trans* fatty acids. However, we did not find that dietary intake had divergent associations with industrial vs.

natural (ruminant) *trans* fatty acid. This is likely because our survey may not have adequately captured the intake of highly processed foods. Additionally, industrial and natural (ruminant) *trans* fatty acids have been reported to have dissimilar associations with some diseases such as coronary heart disease[39], diabetes[40,41], colorectal cancer[42], and pancreatic cancer[43]. However, in our study, both types of *trans* fatty acids showed similar associations with prostate cancer risk and aggressiveness.

Although the dietary intake of fatty acids is a key factor in determining their levels in our body, genetics may also influence fatty acid levels. Multiple genome-wide association studies have shown that polymorphic genetic loci can strongly influence the levels of certain fatty acids in circulation[13,44,45]. We found that SNPs in the *FADS1/2* locus explain up to 19% of the variability of serum arachidonic acid levels in European American men, consistent with prior reports[13]. In contrast, we could not find an association between the same SNPs and arachidonic acid or other omega-6 fatty acids in either African American or Ghanaian men. Previously, a SNP in *FADS1* (rs174548) was shown to be associated with omega-3 fatty acid levels across different ancestries, but data for omega-6 fatty acid levels were not provided by those authors[14]. The lack of association in our study is not explained by a lack of statistical precision in estimation, as the African American and Ghana cohorts were similar sized or even larger than the European American cohort, and two of the studied SNPs (rs174577 and rs174583) did not show large differences in their genotype distribution between the three cohorts (Supplementary Table 6). In contrast, the fatty acid level-associated rs174556 SNP was indeed more common in European American men. Noteworthy, arachidonic acid can be further metabolized by the cyclooxygenase and 5-lipoxygenase enzymes, resulting in the synthesis of pro-inflammatory eicosanoids. One eicosanoid, the pro-metastatic thromboxane A2, has recently been linked by our group to an elevated risk of lethal prostate cancer among African American men[46]. Nonetheless, the *FADS1/2* locus has not been previously linked to prostate cancer in most studies[44].

*Trans* fatty acids have been shown to be linked to systemic inflammation[47,48], including increased levels of inflammatory cytokines such as interleukin (IL)-1β, IL-1, IL-6, tumor necrosis factor 1/2, and monocyte chemoattractant protein 1 in patients with heart failure[49]. In contrast, other studies reported that high intake of *trans* fatty acids may have little or no association with inflammatory markers[50,51]. Chemically and functionally different, omega-3 and omega-6 fatty acids are precursors for several bioactive lipid molecules implicated in immunity and cancer[52–56]. Omega-3 fatty acids derived resolvins have been shown to inhibit angiogenesis[57] and promote termination of tumor promoting inflammation[58]. Therefore, we investigated whether the circulating fatty acid levels that we measured were associated with serum proteome-defined immune-oncological pathways, namely apoptosis/cell killing, autophagy/metabolism, chemotaxis/trafficking to tumor, suppression of tumor immunity (Th2 response, tolerogenic), promotion of tumor immunity (Th1 responses), and vasculature. We found that in our Ghanaian population, omega-6 and omega-3 fatty acids had an overall negative association with the immune-oncological pathways among men without prostate cancer, whereas this association became positive among the men with prostate cancer, along with a robust negative relationship between the omega 6:3 ratio and these same pathways. Animal models and cell culture studies have pointed to a complex mechanism of action between the balance of omega 6 and 3 polyunsaturated fatty acids, and their association with different cancers (including breast and prostate), with sometimes conflicting findings, but overall, studies have shown that omega-3 fatty acids have more so a role in reducing pro-inflammatory cytokines, inflammatory processes, and carcinogenesis than omega-6 fatty acids[59,60]. A negative correlation for omega-3 fatty acids and immune-oncological pathways in cancer cases was seen in the U.S. populations, but the opposite was observed for the Ghanaian population, potentially indicating a population-

specific mechanism based on differences in overall dietary patterns. Additionally, we observed a positive correlation between monounsaturated fatty acids and several immune oncological pathways in European American men. This potentially accounts for the mechanistic role of these fatty acids as regulators of cancer cell death and contributors to cancer migration and invasion due to the conversion of saturated fatty acids to monounsaturated fatty acids by stearoyl-CoA enzyme, which is beneficial to cancer cell survival[61,62]. Because our study observed only a weak negative correlation between *trans* fatty acids and the immune-oncological pathways in cases, any *trans* fatty acid-mediated effects may only partially, if at all, contribute to the positive association between *trans* fatty acids and prostate cancer.

Key strengths of our study include its large sample sizes, the measurement of 24 fatty acids with CLIA-certified technology, and the inclusion of men from Ghana and the U.S., with both African American and European American men in the latter cohort. For the fatty acid measurements, we collected blood samples from all participants. Although blood sample collection in Ghana adhered to a protocol that followed standards of practice in the U.S., serum preparation methods and shipping may have influenced the performance of the fatty acid measurements. Moreover, the fact that only the controls for the Ghana study provided overnight-fasting blood is a limitation of our study. Nevertheless, the differences in fatty acid levels comparing men from the U.S. with men in Ghana in our study were consistent with the differences previously reported for African American men in comparison to Nigerian men[19]. Another limitation encompasses the differences in disease presentation, demographic characteristics, and health indicators between Ghanaian men and men from the U.S. Ghanaian men with prostate cancer were older and presented with more advanced disease, most likely due to the lack of population-based PSA screening for early detection of prostate cancer in Ghana[63]. There is also some evidence that prostate cancer in Ghana may have distinct characteristics[64]. Furthermore, aspirin use was uncommon among Ghanaian men but rather frequent among U.S. men. In contrast to Ghana, aspirin use has been promoted in the U.S. population for prevention of cardiovascular disease. A survey reported an estimated use of aspirin by about 50% of U.S. adults ages 45–75[65], consistent with its use by men in the NCI-Maryland study. Other differences included a significantly higher BMI among U.S. than Ghanaian men that is explained by higher overweight/obesity rates in the U.S. male population at time of recruitment (2005–15) than in the Ghana population (2004–12). The obesity rates in Ghana and other West African countries were reported to be 15–20%[66,67], compared to 34–42% in the U.S. during the same time period[68].

In conclusion, our findings point to a population-specific fatty acid profile that may impact prostate cancer development. Elevated serum *trans* fatty acid levels were associated with higher odds of prostate cancer in the Ghanaian, African American, and European American men, thereby identifying *trans* fatty acids as potential prostate cancer risk factors, independent of ancestry or geographic location. Although the extent to which fatty acids are prostate cancer risk factors remains controversial, the relationship between fatty acids, immune function, and prostate cancer, specifically in men of African descent, merits further exploration to determine causal relationships.

## Methods

### NCI-Maryland prostate cancer case-control study
This study was designed to examine the contribution of environmental exposures and ancestry-related factors to the excessive prostate cancer burden among African American men in the Baltimore metropolitan region[9,10]. The study protocol was approved by the NCI(protocol # 05-C-N021) and the University of Maryland (protocol #0298229) Institutional Review Boards. The study complies with all relevant ethical regulations, and all participants signed an informed consent. Participants in the study self-reported as male gender in the study

questionnaire when given the choice between male and female gender. Given the focus of this study is on prostate cancer, the study design was restricted to only individuals who indicated male on their study questionnaire and are referred to as men in the manuscript. Disaggregated sex and gender data were not collected for this study. Men with prostate cancer were recruited at the Baltimore Veterans Affairs Medical Center and the University of Maryland Medical Center. A total of 976 cases (489 African American and 487 European American men) were enrolled into this study between 2005 and 2015. Controls were identified through the Maryland Department of Motor Vehicle Administration database and were frequency-matched to cases on age and self-reported race. A total of 1033 population controls were recruited (485 African American and 548 European American men). At the time of enrollment, both cases and controls were administered a survey by a trained interviewer and a blood sample was collected. Participants in this study were compensated up to 50 USD for their participation in this study. Serum samples were available for 846 cases (407 African American and 439 European American) and 845 controls (381 African American and 464 European American), and therefore only these individuals contributed to this study. Most of the 846 cases (85%) were enrolled within a year of the disease diagnosis, with a median of 5.1 months between disease diagnosis and blood collection.

### NCI-Ghana prostate cancer case-control study

This case-control study was designed to study lifestyle, environmental, and genetic risk factors for prostate cancer in indigenous African men[11]. The study was approved by Institutional Review Boards at the University of Ghana (protocol #001/01-02) and the NCI (protocol #02CN240). Prior to study enrollment, all participants signed an informed consent, and the study complies with all relevant ethical considerations. Men with prostate cancer were recruited at Korle Bu Teaching Hospital in Accra, Ghana between 2008 and 2012. The cases were diagnosed using Digital Rectal Exam (DRE) and PSA tests, followed by biopsy confirmation. Immediately after diagnosis and before treatment, cases were consented and asked to submit a blood specimen and questionnaire data. Controls were identified through probability sampling using the 2000 Ghana Population and Housing Census data to recruit approximately 1000 men aged 50–74 years in the Greater Accra region between 2004 and 2006. These men were confirmed to not have prostate cancer by PSA testing and DRE. Participants were compensated up to 5 USD for transportation costs related to the study. Serum samples were available for 585 prostate cancer cases and 658 population controls; hence, only these individuals were used for the study herein.

### Serum sample processing

The participants in the two studies provided blood samples at time of recruitment. For the NCI-Maryland study, most blood samples were processed the same day, but always within 48 h, after storage in a refrigerator. Both cases and controls provided non-fasting blood. For the NCI-Ghana study, blood samples were processed within 6 h. In this study, controls were asked to provide overnight-fasting blood while cases were not asked to provide non-fasting blood. Serum was prepared using standard procedures and aliquots were stored at −80 °C. Serum samples were shipped from Ghana to the NCI in dry ice boxes.

### Serum fatty acid measurement

Absolute (μg/mL) serum concentrations of 24 fatty acids were measured using gas chromatography (GC) with flame ionization detection by a CLIA-certified laboratory, OmegaQuant Analytics, using the following procedure. Serum (and an internal standard) was transferred to a screw-cap glass vial, dried down with a speed-vac and BTM (methanol containing 14% boron trifluoride, toluene, methanol; 35:30:35 v/v/v) (Sigma-Aldrich, St. Louis, MO) was added. The vial was briefly vortexed and heated in a hot bath at 100 °C for 45 min. After cooling, hexane (EMD Chemicals, USA) and HPLC grade water was added, the tubes

were recapped, vortexed and centrifuged help to separate layers. An aliquot of the hexane layer was transferred to a GC vial. GC was carried out using a GC-2010 Gas Chromatograph (Shimadzu Corporation, Columbia, MD) equipped with a SP-2560, 100-m fused silica capillary column (0.25 mm internal diameter, 0.2 um film thickness (Supelco, Bellefonte, PA). Fatty acids were identified by comparison with a standard mixture of fatty acids (GLC OQ-A, NuCheckPrep, Elysian, MN) which was also used to determine individual fatty acid calibration curves. The following 24 fatty acids (by class) were identified: saturated (14:0, 16:0, 18:0, 20:0, 22:0 24:0); *trans* (16:1n-7t, 18:1n-9t, 18:2n-6t, 9t); *cis* monounsaturated (16:1n-7, 18:1n-9, 20:1n-9, 24:1n-9); *cis* n-6 polyunsaturated or omega-6 (18:2n-6, 18:3n-6, 20:2n-9, 20:3n-6, 20:4n-6, 22:4n-9, 22:5n-9); and *cis* n-3 polyunsaturated or omega-3 (18:3n-3ccc, 20:5n-3, 22:5n-3, 22:6n-3) (Supplementary Table 2). The chromatographic conditions used in this study were sufficient to isolate the C16:1 *trans* isomers and the C18:2 Δ9t-12c, 9t-12t, and 9c-12t isomers, which was reported as C18:2n6t. However, each individual C18:1 *trans* molecular species (i.e., C18:1 Δ6 thru Δ13) could not be segregated but appeared as two blended peaks that eluted just before oleic acid. The areas of these two peaks were summed and referred to as C18:1 *trans*. The serum samples from the NCI-MD study (846 cases and 845 controls) and the NCI-Ghana study (585 cases and 658 controls) were randomized and assayed in that order. In addition to the built-in internal controls, 5% blinded duplicates were randomly selected and were randomized along with the original set of samples. The median Coefficient of Variation (CV) calculated based on the 156 blind duplicates was 5.1% across the 24 fatty acids where the CVs among duplicates for 20 out of the 24 markers were <15% (Supplementary Table 16). Fatty acids were assessed individually as well as grouped into five distinct chemical classes/structural groups: saturated, *trans*, *cis*-monounsaturated, omega-3, and omega-6 fatty acids (Supplementary Table 2). OmegaQuant Analytics provided measurements for the 24 fatty acids and assigned them a classification (saturated, *trans*, etc.). We created an abundance level score for each class by adding together concentrations of individual fatty acids that were grouped together (e.g., *trans* fatty acid class includes the combined concentrations for elaidic, palmitelaidic, and linoelaidic acids). Additionally, the omega 6:3 ratio, an important indicator of dietary health[12,69], was included in the fatty acid analysis.

### Analysis of the association between clinical/socio-demographic characteristics and circulating fatty acids

As done previously in a proteomic study of these cohorts[4], we assessed the associations of age, BMI, education, smoking, diabetes, and aspirin use with the abundance of individual circulating fatty acids (as continuous values) by means of multivariable linear regression models implemented by the function lm in the base R package stats (version 3.6.1). For each fatty acid, we fitted the formula "fatty acid ~ age + bmi + education + smoking + diabetes + aspirin", which yielded the model's F-statistic and associated F-statistic *p*-value, as well as the intercept and regression coefficients with their associated standard errors (SE) and P values. F-statistic P-values were adjusted by FDR across all models; moreover, within each model, regression coefficient P values were also FDR-adjusted. Full regression results for each cohort are provided as Supplementary Data 1. A fatty acid (as response variable) was considered significantly associated with clinical and socio-demographic covariables if the multivariable model yielded an FDR-adjusted P-value < 0.05 on the F-statistic. If this condition was satisfied, the association between the target fatty acid and each individual covariable was characterized by the corresponding FDR-adjusted P-value and coefficient.

### Serum protein measurement

Serum levels of 82 immuno-oncology proteins were measured simultaneously using a proprietary multiplex Proximal Extension Assay

Article

(PEA) (i.e. Olink IMMUNO-ONCOLOGY panel v.3101) by Olink Proteomics (Boston)[10]. Serum samples from the NCI-MD study (846 cases and 845 controls) and NCI-Ghana study (585 cases and 658 controls) were randomized and were assayed in that order. 5% blinded duplicates were randomly selected and were randomized along with the original set of samples. Olink utilizes a relative quantification unit, Normalized Protein eXpression (NPX), which is in a $\log_2$-format. Protein levels were intensity normalized to adjust for batch effects. Ninety-five percent of the samples passed a stringent quality control (NCI-MD study: 819 cases and 828 controls; NCI-Ghana study: 489 cases and 654 controls) – with coefficients of variation (CV) among duplicates at <10% for every marker.

### Functional annotation of serum proteins into pathway and pathway activity scores

Proteins were grouped into six biological processes (i.e., apoptosis/cell killing, autophagy/metabolism, chemotaxis/trafficking to tumor, suppression of tumor immunity (Th2 response, tolerogenic), promotion of tumor immunity (Th1 responses), and vasculature[10]. A description of the 82 immuno-oncology proteins and their assignment to pathways can be found in Supplementary Table 17. Apoptosis, autophagy, chemotaxis, suppression of tumor immunity, promotion of tumor immunity, or vasculature activity scores were calculated for each study participant as the mean z-score value for the proteins belonging to the respective biological process with higher scores indicating a higher pathway activity in an individual. The association between each immune-oncological biological pathway and fatty acid class were evaluated using pairwise linear correlation tests.

### Analysis of variance

Variance analysis for the levels of each of the 24 fatty acids were simultaneously assessed as a function of demographic, clinical, and genetic factors in men with prostate cancer from the NCI-Maryland and NCI-Ghana studies. Data on dietary intake was available only for the participatnts of the NCI-Maryland study. The analyses were implemented by the function aov in the base stats R package (version 3.6.1).

### FADS1/2 locus and circulating fatty acid levels

We assessed the association of germline genotypes in the fatty acid desaturase 1 and 2 locus (FADS1, rs174556; FADS2, rs174577 and rs174583) with the levels of the 24 individual fatty acids. All genotyping data was generated for the NCI-Maryland and NCI-Ghana studies[11] at the Cancer Genomics Research Laboratory/NCI-Leidos, a genotyping core facility within NCI-DCEG. All samples passed stringent quality control measures for GWAS data. SNP genotype data was generated using the Infinium HumanOmniS-Quad BeadChip array and analyzed using the High-Throughput Workflow section on the standard Illumina microarray data analysis workflow page. Genotype calls initially generated as Genotype Call files (.gtc) were converted to PLINK 1.9 using the Illumina-provided open-source library (github.com/Illumina/BeadArrayFiles). FADS1/2 genotypes were selected based on both coverage of the FADS1/2 locus by the array and literature reports showing that this genetic locus and the genotypes influence circulating fatty acid levels in individuals of European descent[13,14]. More information about the selected SNPs and their frequency by population group can be found in Supplementary Table 6.

### Unsupervised hierarchical clustering of fatty acids

Heatmaps and dendrograms showing unsupervised hierarchical clustering of 24 individual fatty acids were generated using JMP 14.0.

### Statistical analysis

Data analyses were performed using Stata/SE 16.0, JMP 14.0, and R statistical packages. All statistical tests were two-sided. An association was considered statistically significant with $P < 0.05$ or Bonferroni-corrected significance threshold in instances where correction for multiple testing was required. Student's t-tests were used to compare fatty acid mean concentrations by population group. Unconditional logistic regression was used to compute the odds ratios (OR) and 95% confidence intervals (CI) to assess the association of circulating levels of fatty acids with prostate cancer. All tests used continuous fatty acid concentration data unless otherwise noted. We adjusted for potential confounding factors: age at recruitment (years), body mass index at study enrollment (BMI, kg/m$^2$), education (high school or less, some college, college, professional school), diabetes(yes/no), smoking history (never, former, current), and aspirin use (regular user, yes/no) where appropriate.

### Ethics & inclusion statement

This mutli-regional study was conducted in collaboration with local researchers in Ghana and was approved by the local ethics review committee at the University of Ghana (protocol #001/01-02).The researchers were involved in the research process including study design, study implementation and authorship of publications. NCI-Ghana Prostate Cancer Case-Control was designed to study lifestyle, environmental, and genetic risk factors for prostate cancer in indigenous African men. Hence, findings from the study can be used to address the disproportionately high prostate cancer burden in the region.

### Reporting summary

Further information on research design is available in the Nature Portfolio Reporting Summary linked to this article.

## Data availability

Clinical, demographic and molecular data used for this study (i.e., self-reported race, age, BMI, education, aspirin use, diabetes, smoking status, NCCN risk score, proteomics data, GWAS data, and fatty acid data) are deposited in the Open Science Framework database (https://osf.io/tscgh/) under the accession code TSCGH[70] and as a public GitHub repository at https://github.com/tsionzminas/Prostate-Cancer-and-Circulating-Fatty-acids or at Zenodo under the accession code 8023186[71]. The full proteomics data was deposited in the Open Science Framework database under the accession code 327HA[72]. Source data are provided with this paper. The remaining data are available within the paper, Supplementary Information, and Supplementary Data. Source data are provided with this paper.

## Code availability

The scripts used in our bioinformatics pipeline to perform data analysis and visualization are available as a public GitHub repository at https://github.com/tsionzminas/Prostate-Cancer-and-Circulating-Fatty-acids or at Zenodo under the accession code 8023186[71].

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

## Acknowledgements
First and foremost, we would like to extend our sincere gratitude to the study volunteers of both the NCI-Ghana and NCI-Maryland Prostate Cancer Case-Control studies for their invaluable contribution. We would also like to thank personnel at the University of Maryland and the Baltimore Veterans Administration Hospital for their contributions with the recruitment of participants into the NCI-Maryland study. We would further like to thank Prof. Edward D. Yeboah as the original Ghana PI and Ms. Evelyn Tay as the original Study Manager for the NCI-Ghana study. We would also like to thank our funders: DoD award W81XWH1810588 (SA, CY). U54 CA118623(NCI) and U54-MD007585-26-CY (NIMHD) (both to CY). NCI Intramural Research Program, Center for Cancer Research (SA) and Division of Cancer Epidemiology and Genetics (MBC). Cancer Research Training Award (TZM) and NCI Cancer Prevention Fellowship Program (BDL).

## Author contributions
Conceptualization: T.Z.M., B.D.L., C.Y., M.B.C., S.A.; Data curation: T.Z.M., T.H.D., M.B.W., C.J.S., S.V.J., O.M.O., A.A., J.L.; Formal Analysis: T.Z.M., B.D.L., A.L.Z., J.C.; Funding acquisition: C.Y., M.B.C., S.A.; Investigation: T.Z.M., B.D.L.; Methodology: T.Z.M., B.D.L., A.L.Z., W.T., J.C., J.L., C.A.L., M.B.C., S.A.; Project administration: T.H.D., F.B., A.A.; Resources: W.T., Y.T., R.B.B., A.A.A., J.E.M, R.N.H., A.W.H., MBC, S.A. Supervision: S.A.; Visualization: T.Z.M, B.D.L., J.C.; Writing – original draft: T.Z.M., B.D.L.; Writing – review & editing: T.Z.M., B.D.L., J.C., F.B., M.B.W., W.T., C.A.L., S.A.

## Funding

## Competing interests
C.Y. received honorarium or consultant fees from PreludeDx, QED Therapeutics, Amgen, Regeneron, and Riptide Biosciences. C.Y. is a shareholder in Riptide Biosciences. The remaining authors declare that they have no competing interests.
