## [Peer Review File · Nature Communications]

REVIEWER COMMENTS

Reviewer #1 (Remarks to the Author):

The authors have undertaken a suitably powered and careful case-control evaluation of circulating fatty acid levels and of serum proteome-defined pathways in Ghanaian, African-American (AA) and European American (EA) men focussing on associations with prostate cancer risk. The data are clearly presented, the conclusions are well-justified and the discussion is well-balanced. I would like to see commentary in the latter section on the limitations of the study. As one might expect the data for Ghanaian case-controls shows very different age patterns, patterns of aspirin usage and so forth than the AA and EA groups. Little attention is drawn to this in the text and to its implications. A median age of 59 amongst Ghanaian controls and a median age of 70 amongst cases suggests that the pathway for prostate cancer diagnosis may be rather different versus the AA and EA groups where median ages for controls and cases lie in the range 63-66.5. This in turn is reflected in the Gleason scores in the case groups - there are over twice as many high Gleason score cases (8-10 as reported here) in the Ghanaian group versus the AA and EA groups. PSA levels are also 10 times higher. This suggests that the disease may be very different at diagnosis and consequently one might expect differences in the blood measurements to track with this. Much more could be made of this in the text and perhaps even in the figures. For example, does the case clustering in Figure 1b track with the Gleason pathology amongst the detected cases? I note that some of the Ghanaian cases cluster with EA and AA cases and a substantial proportion form a distinct group. That obviously is driven in Figure 1b by circulating fatty acid levels. However does that then result, for example, in intermediate Grade cases amongst the Ghanaian group clustering with EA/AA cases? This is a valuable and interesting study that will be of great interest to the field.

Reviewer #2 (Remarks to the Author):

This manuscript reports a study of the circulating fatty acid levels related to population groups and prostate cancer. As stated by the authors themselves, the strength of this work is the systematic study with a large sample size. Samples from 2934 men were used, including Ghana, African and European Americans from the United States. CLIA-certified technologies were used for the sample analysis.

This type of studies would certainly be meaningful for identifying biomarkers for diseases or shedding light on fundamental studies of metabolic pathways. The findings for this particular study, however, might not justify a significant advancement in knowledge or method. Relatively apparent differences were observed between populations, while the differences between the control and prostate cancer cases are insignificant. The differences found, such as omega-3 fatty acid concentrations higher in Ghanaian and trans fatty acids increase odds for prostate cancer cases, might be attributed to differences in eating habit, which have been previously reported.

Verification with fundamental biological studies could certainly help to improve the depth of this study but might need a quite amount of additional work.

The following suggestions might help, but only within the current scope of the study.

One way to improve the story, might be shifting the emphases away from “link to prostate cancer”, with a focus on the variations among population groups, and a minor mentioning of the indications to prostate cancer.

Major concerns:

1. In addition to the differences reported for types of fatty acids, variants related to specific fatty acids could also be reported, in order to provide values to study of the metabolic changes.
2. The changes found in the serum samples in this work could be further explored, with verification studies involving prostate cancer cell lines or prostate cancer tissues. For better explaining the differences, the range of metabolites could also be expanded, including phospholipids and glycerides.
3. The correlations of immune-oncological markers and fatty acids were shown based on data analysis, while there was no evidence that these fatty acids were involved in related pathways. Relevant supporting information needs to be added.
4. In addition, there was no investigation for comparing the differences of pathway activities among population groups, which might be closely relevant to the findings reported in this manuscript.

Other Comments:

1. Figure 1c-e can be rearranged to show better the comparison among the population groups, which is the focus of the discussion for this part of the main text.
2. Figure 3 and 4 could be combined to illustrate the genetic effects.
3. In Figure 6 showing the heatmap patterns of correlation between immune-oncological pathway activity and circulating fatty acid level, significant difference between control and cancer cases was shown but was not explained in the main text.

Reviewer #3 (Remarks to the Author):

Authors determined the levels of 24 fatty acids in serum samples of prostate cancer patients (PCA) and healthy volunteers (control) from Ghana and United States (African American and European American were represented at similar number). This allowed the authors to stratify the patients into three ethnic categories: Ghanaian, African American and European American.

The main finding is that American men show higher levels of trans fatty acids, and the risk of developing prostate cancer is correlated with elevated trans fatty acid levels. In other words, they confirmed a commonly accepted information that trans fatty acid high levels correlate with the risk of prostate cancer.

They also found that in Ghana men, omega-3 fatty acids are present at higher levels than in American men, regardless of their African or European ancestry. Omega-3 fatty acids are considered to show a potentially protective effect against prostate cancer [Cancer Epidemiol. Biomark. Prev. 2013, 22, 697–707].

Main concerns:

The title is not informative.

How the samples collection was planned is not explained in sufficient terms. The authors state: “Only population controls provided overnight-fasting blood”.

Also taking the fasting/feeding status into account, comparing individual fatty acid levels in non-fasted prostate cancer patients with fasted healthy men (especially in fig 5. – volcano plot) seems risky

Additionally, Fig2. Presents multiple factors that did (or did not) correlate with specific fatty acids with relation to age, BMI, education, aspirin use, smoking or diabetes, but NO info on how advanced the disease was. There is no information in the paper about stratification of patients according to i.e. Gleason score, at the moment of diagnosis/sample collection. Including this information would be a valuable addition to analysis, if the data are available.

Fig. 3 and Fig.4. are unclear to me. Fatty acid desaturases (encoded by the FADS1 and FADS2 genes) are enzymes involved in the synthesis of polyunsaturated fatty acids, namely in the conversion of linoleic acid into arachidonic acid (AA), and of α -linolenic acid (ALA) into eicosapentaenoic acid EPA and docosahexaenoic acid. Why is there in the fig 3 and 4 the whole list of fatty acids, including those that are not supposed to be metabolized by these enzymes, instead of the relevant substrates and products only?

Reviewer #4 (Remarks to the Author):

I would like to thank the authors for this important paper. This paper is important when considering the sample size, the population structure, and the quality of the analysis conducted.

MAJOR COMMENTS

- My main concern was with regards to the units of the fatty acids. Although I totally agree with the use of absolute vs. relative concentrations, it does not make much of a sense when concentrations are summed up. For example, what does the total of all fatty acids stand for? I would recommend that the authors keep the sub-total but remove the grand total of the fatty acids.
- I suspected that the authors presented OR for prostate cancer by unit increment. This is not correct and is likely explaining results such as OR and 95%CI of 1.00 (1.00-1.00) $P < 0.001$ (table 2). The results in Table 2 should be rather presented for log-transformed values of the fatty acids, or (I would prefer) by standard deviation increment for each fatty acid.
- Regarding trans fatty, I would strongly recommend to the authors to further consider the subdivision, ruminant trans and industrial trans for two main reasons. We now know that both industrial and ruminant fatty acids have divergent dietary sources, and often dissimilar associations with diseases.
- Overall, the discussion section needs to be improved. First, the authors did not comment on the inverse association observed with EPA, DPA, and DHA. This is equally an important finding and may allude to a possible beneficial effect of fish consumption? Second, the authors should comment separately on elaidic acid and palmitelaidic acid. Elaidic acid is a marker of processed food intake (PMID: 25675445) and is considered as an industrial trans whereas palmitelaidic is a ruminant trans fatty acid (PMID: 33913149).
- The authors should provide the full nomenclature of the fatty acids, especially in Supp Table 2 and appropriately within the manuscript from line 386 (see PMID: 28465289)

MINOR COMMENTS

About some syntaxes

- It is preferred to state that a risk factor is associated with high or low risk, not "increased" or "decreased" risk, which suggests a referential start for the risk.
- The authors routinely used "yet" and "still" especially at the beginning of sentences, although the link of these adverbs with preceding affirmations were unclear. Lines 51, 60, 163, 238, 253...should be re-arranged
- The authors should avoid some generic sentences, which in my opinion were misleading. Line 284: "Fatty acid levels are under genetic control". This is not necessarily true. Many fatty acids are directly sourced from the diet, and their circulating levels confirm past intakes. Another generic sentence: "Our study has strengths and limitations". This is an empty sentence.

ABSTRACT

- The first sentence is utterly wrong. It assumed a positive association for a group of molecules with disparate structures and functions. The association between fatty acids and cancers is specific for each fatty acid and cancer site.
- Line 45: replace examined by "analysed", the sentence is referring to the laboratory work.
- Line 51: please make that sentence clear and straightforward. Just state the associations between TFA and prostate cancer.

INTRODUCTION

- Line 69-74: there is a need to improve that section. The authors rightly cited two meta-analyses that investigated the relationships between fatty acids and prostate cancer, one on dietary fatty acids and a second on circulating levels. Then, the authors went on to elaborate on a result of a case-cohort study. This looks like cherry-picking to me. The authors should rather comment on the meta-analyses, especially on circulating levels, and succinctly present known associations with prostate cancer.

Line 78: It is preferable to say aim rather than goal

Line 78-88: This is confusing. Are these the objectives? It sounds like results to me

RESULTS

Line 103: I would not say "global", you mean substantial?

TABLE 1

- Provide the number of cases/controls on top of the table
- By age of study entry, you mean age at recruitment? These are case-control studies, we mostly talk about entry for cohorts. Please correct throughout the manuscript

TABLE 2

- How are the tertiles created? Normally in case-control studies, the quantiles are created based on the controls, then transferred to the cases.
- It is not common practice to provide % of cases in such tables.
- No need to provide P value, only P for trend is important. And it should be stated "P trend", not just trend.
- Median values of the concentration for each tertile should be provided

METHODS

- Line 396: why "completely" randomised, please correct that

SUPPLEMENTARY TABLE 1

Why AA and EA below diabetes? Maybe this is just an issue with the preparation of the pdf?

DISCUSSION

- Line 254: Please check the paper in Uganda (PMID: 30477593) and in Asian Indians (PMID: 27374582) to discuss the sources of trans fatty acids in other populations

We thank the reviewers for their insightful comments. We provide the reviewers' comments together with our responses below. We show edits in the manuscript using track changes.

Reviewer # 1

Comment #1: *I would like to see some commentary in the latter section on the limitations of the study. As one might expect the data for Ghanaian case-controls shows very different age patterns, patterns of aspirin usage and so forth than the AA and EA groups. Little attention is drawn to this in the text and to its implications. A median age of 59 amongst Ghanaian controls and a median age of 70 amongst cases suggests that the pathway for prostate cancer diagnosis may be rather different versus the AA and EA groups where median ages for controls and cases lie in the range 63-66.5.*

Response: Thank you for raising this point. We agree with the reviewer and now added a paragraph to the limitation section that addresses these concerns. On pages 16-17, it reads as follows:

“Another limitation encompasses the differences in disease presentation, demographic characteristics and health indicators between the Ghanaian and US men. Ghanaian men with prostate cancer were older and presented with more advanced disease, most likely due to the lack of population-based PSA screening for early detection of prostate cancer in Ghana⁶³. Additionally, there is some evidence that prostate cancer in Ghana may have distinct characteristics⁶⁴. Aspirin use was uncommon among Ghanaian men but rather frequent among US men. In contrast to Ghana, aspirin use has been promoted in the U.S. population for prevention of cardiovascular disease. A survey reported an estimated use of aspirin by about 50% of U.S. adults ages 45-75⁶⁵, consistent with its use by men in the NCI-Maryland study. Other differences included a significantly higher BMI among U.S. than Ghanaian men that is likely explained by higher overweight/obesity rates in the US male population at time of recruitment (2005-15) than in the Ghana population (2004-12). The obesity rates in Ghana and other West African countries were reported to be 15-20%^{66,67}, compared to 34-42% in the U.S. during the same time period⁶⁸.”

Comment #2: *This in turn is reflected in the Gleason scores in the case groups - there are over twice as many high Gleason score cases (8-10 as reported here) in the Ghanaian group versus the AA and EA groups. PSA levels are also 10 times higher. This suggests that the disease may be very different at diagnosis and consequently one might expect differences in the blood measurements to track with this. Much more could be made of this in the text and perhaps even in the figures. For example, does the case clustering in Figure 1b track with the Gleason pathology amongst the detected cases? I note that some of the Ghanaian cases cluster with EA and AA cases and a substantial proportion form a distinct group. That obviously is driven in Figure 1b by circulating fatty acid levels. However does that then result, for example, in intermediate Grade cases amongst the Ghanaian group clustering with EA/AA cases? This is a valuable and interesting study that will be of great interest to the field.*

Response: We agree with the reviewer that disease presentation is different between the Ghanaian and U.S. men with prostate cancer, with significantly more Ghanaian men presenting with high grade disease. Generally, prostate cancer is more advanced in the Ghanaian men, mainly because an early diagnosis is very uncommon in Ghana but common in the U.S., with PSA screening rates being about 70% for men in Maryland during the recruitment period. In addition, there could be an intrinsically more aggressive disease affecting men in Ghana, which is still an unresolved research question. We addressed this in our response to comment #1.

To investigate the reviewer's concern we performed additional analyses seeking to define the relationship between circulating fatty acids and Gleason sum scores. In our analysis we focused on the contrast Gleason sum score ≤ 7 versus ≥ 8 because the major difference between Ghanaian and US men with prostate cancer is the increased frequency of Gleason sum score ≥ 8 . We added this dichotomized Gleason score to the hierarchical clustering heatmap in Suppl. Fig. 1 and noted in the manuscript text that we did not see a particular clustering by Gleason score. In a more quantitative approach to see if Gleason score significantly associates with fatty acid levels, we performed a multivariable linear regression analysis after adding Gleason score as a covariate along with age, BMI, education, smoking, diabetes, and aspirin for each population group. The main findings are summarized in the Supplementary Data 2 file. We did not find that there are significant associations between Gleason score and the circulating levels of the 24 fatty acids in the three groups of men with prostate cancer, Ghanaian, African-American, and European-American men.

In the revised manuscript text, on pages 6-7, we wrote:

*"In an analysis restricted to cases, we explored possible associations between Gleason score and the 24 individual fatty acids using the same multivariable linear regression model but with addition of Gleason score as a covariate, dichotomized as ≤ 7 versus ≥ 8 . In this analysis across the three patient groups, Gleason score did not associate significantly with any of the fatty acids analyzed in this study (**Supplementary Data 2**)."*

Reviewer #2

Comment: *This manuscript reports a study of the circulating fatty acid levels related to population groups and prostate cancer. As stated by the authors themselves, the strength of this work is the systematic study with a large sample size. Samples from 2934 men were used, including Ghana, African and European Americans from the United States. CLIA-certified technologies were used for the sample analysis. This type of studies would certainly be meaningful for identifying biomarkers for diseases or shedding light on fundamental studies of metabolic pathways. The findings for this particular study, however, might not justify a significant advancement in knowledge or method. Relatively apparent differences were observed between populations, while the differences between the control and prostate cancer cases are insignificant. The differences found, such as omega-3 fatty acid concentrations higher in Ghanaian and trans fatty acids increase odds for prostate cancer cases, might be attributed to differences in eating habit, which have been previously reported. Verification with*

fundamental biological studies could certainly help to improve the depth of this study but might need a quite amount of additional work. The following suggestions might help, but only within the current scope of the study.

Comment #1: *One way to improve the story, might be shifting the emphases away from “link to prostate cancer”, with a focus on the variations among population groups, and a minor mentioning of the indications to prostate cancer.*

Response: Thank you for the suggestion. The described work has been funded by a Prostate Cancer DoD Impact award to study candidate causes of prostate cancer in African descent men. As such, the work and manuscript would need a focus on prostate cancer. Also, we thought our observations that all measured trans fatty acids associated with increased odds of prostate cancer, independent of ancestry or geographic location, is a key observation. Thus, we would prefer not to change the focus of the manuscript away from prostate cancer.

Comment #2: *In addition to the differences reported for types of fatty acids, variants related to specific fatty acids could also be reported, in order to provide values to study of the metabolic changes.*

Response: We agree with this reviewer’s comments that in addition to the data on population differences in fatty acid types (classes) as reported in Figure 1C-E, differences in individual fatty acids would be an important data for the readers. We provide those data in Supplementary Table 3. However, we are reluctant to describe all of these data points in the current manuscript. We believe it would make the Results and Discussion unfocused – just too much information. Here, we want to point out that all source data will be made publicly available. Others are welcome to further analyze the data.

Comment #3: *The changes found in the serum samples in this work could be further explored, with verification studies involving prostate cancer cell lines or prostate cancer tissues. For better explaining the differences, the range of metabolites could also be expanded, including phospholipids and glycerides.*

Response: Thanks. We cannot easily study the metabolites in prostate tumors as frozen tumor tissue needed for the analysis is not available – or if available for only few cases but not from Ghana. To study them in tumor tissue it would need access to tumor cores – we don’t have them. We cannot not see how working with prostate cancer cell lines would help verifying our data. Many of these fatty acids are not produced by the cancer cells and studying population differences would be difficult to impossible. If one would explore the role of fatty acids in prostate cancer, animal models would be preferred but those are beyond the scope of this manuscript. We would not be able to measure additional fatty acids. The 24 fatty acids measured for this study were measured with assays for clinical practice by OmegaQuant, and are CLIA-certified. They come as a pre-specified package, focusing on fatty acids that are thought to be important indicators for human health. We think that the list of 24 measured fatty acids is already quite comprehensive.

However, to address your point more directly, there is literature on the effects of *trans* fatty acids in animal and cell culture cancer models. We now added some of this literature to the manuscript body. On page 12, discussion, we write as follows:

“Previous studies in experimental systems have shown that trans fatty acid may have pro-tumorigenic properties²⁰⁻²³, providing plausibility for observations from cancer epidemiology linking trans fatty acids to several human cancers⁶.”

Comment #4: *The correlations of immune-oncological markers and fatty acids were shown based on data analysis, while there was no evidence that these fatty acids were involved in related pathways. Relevant supporting information needs to be added.*

Response: We thank the reviewer for this comment. We expanded the discussion of our findings in the manuscript. In the revised manuscript, on page 15, we write as follows:

“Trans fatty acids have been shown to be linked to systemic inflammation^{47,48}, including increased levels of inflammatory cytokines such as interleukin (IL)-1 β , IL-1, IL-6, tumor necrosis factor 1/2, and monocyte chemoattractant protein 1 in patients with heart failure⁴⁹. In contrast, other studies reported that high intake of trans fatty acids may have little or no association with inflammatory markers^{50,51}. Chemically and functionally different, omega-3 and omega-6 fatty acids are precursors for a number of bioactive lipid molecules implicated in immunity and cancer⁵²⁻⁵⁶. Omega-3 fatty acids derived resolvins have been shown to inhibit angiogenesis⁵⁷ and promote termination of tumor promoting inflammation⁵⁸. Therefore, we investigated whether the circulating fatty acid levels that we measured were associated with serum proteome-defined immune-oncological pathways, namely apoptosis/cell killing, autophagy/metabolism, chemotaxis/trafficking to tumor, suppression of tumor immunity (Th2 response, tolerogenic), promotion of tumor immunity (Th1 responses), and vasculature. We found that in our Ghanaian population, omega-6 and omega-3 fatty acids had an overall negative association with the immune-oncological pathways among men without prostate cancer, whereas this association became positive among the men with prostate cancer, along with a robust negative relationship between the omega 6:3 ratio and these same pathways. Animal models and cell culture studies have pointed to a complex mechanism of action between the balance of omega 6 and 3 polyunsaturated fatty acids (PUFAs), and their association with different cancers (including breast and prostate), with sometimes conflicting findings, but overall, studies have shown that omega-3 PUFAs have more so a role in reducing pro-inflammatory cytokines, inflammatory processes, and carcinogenesis than omega-6 PUFAs^{59,60}. A negative correlation for omega-3 PUFAs and immune-oncological pathways in cancer cases was seen in the US population, but the opposite was observed for the Ghanaian population, potentially indicating a population-specific mechanism based on differences in overall dietary patterns. Additionally, we observed a positive correlation between monounsaturated fatty acids and several immune oncological pathways in EA men. This potentially accounts for the mechanistic role of these fatty acids as regulators of cancer cell death and contributors to cancer migration and invasion due to the conversion of saturated fatty acids to monounsaturated fatty acids by stearoyl-CoA enzyme, which is beneficial to cancer cell survival^{61,62}. Because our study observed only a weak negative correlation between trans fatty acids and the immune-oncological pathways in cases, any trans

fatty acid-mediated effects may only partially, if at all, contribute to the positive association between trans fatty acids and prostate cancer.”

Comment #5: In addition, there was no investigation for comparing the differences of pathway activities among population groups, which might be closely relevant to the findings reported in this manuscript.

Response: We agree with the reviewer that this stratification was missing. Indeed, the serum proteome-based immune-oncological pathway activities vary by population groups, as reported by us in our previous publication (PMID: 35365620). Therefore, we have now revised previous Fig. 6 (now Fig. 5) to show additional correlation matrices according to population group.

Comment #6: Figure 1c-e can be rearranged to show better the comparison among the population groups, which is the focus of the discussion for this part of the main text.

Response: Thank you. However, we find it hard to re-arrange this figure, and kept the original version in the submitted revision. It comprehensively captures the data, by displaying ratios.

Comment #7: Figure 3 and 4 could be combined to illustrate the genetic effects.

Response: Per the recommendation, we have combined Figures 3 and 4 into one revised figure, which is now Fig. 3.

Comment #8: In Figure 6 showing the heatmap patterns of correlation between immune-oncological pathway activity and circulating fatty acid level, significant difference between control and cancer cases was shown but was not explained in the main text.

Response: Thank you for pointing this out. We now added text to the manuscript that describes the findings captured by this figure, now Figure 5 in the revised manuscript.

On pages 9-10 of the manuscript, we write as follows:

“Associations of circulating fatty acids with an immune-oncology marker profile

Circulating fatty acids may influence the immune environment. For the NCI-Maryland and NCI-Ghana studies, 82 immune-oncological markers have previously been measured and grouped into pathways: apoptosis/cell killing, autophagy/metabolism, chemotaxis/trafficking to tumor, suppression of tumor immunity (Th2 response, tolerogenic), promotion of tumor immunity (Th1 responses), vasculature⁴. Thus, we assessed whether there was a relationship between the immune-oncology marker-defined pathways and circulating fatty acid levels. For this analysis, we grouped the fatty acids into the five classes as already described and then assessed the relationships between fatty acid classes and pathway activity scores (see methods) by population group and for men with and without prostate cancer. As shown in Fig. 5a, the correlation heatmaps covering the controls revealed significant negative associations between the omega-6 fatty acid class and several of the immune-oncological pathways, as well as a positive association between monounsaturated fatty acids and the vasculature pathway, but only among Ghanaian men (Fig. 5a, Bonferroni-corrected $P < 0.01$). In Ghanaian men with prostate cancer, there were significant positive associations between the omega-3 and omega-6 fatty acid classes and the immune-oncological pathways, and significant negative associations

between the omega 6:3 ratio and the activity scores for these pathways (Fig. 5b, Bonferroni-corrected $P < 0.01$). For EA men with prostate cancer, we observed significant positive associations between monounsaturated fatty acids and the apoptosis, inflammation, suppression of tumor immunity and vasculature pathways. In addition, saturated fatty acids associated similarly with the pathway activity score for suppression of tumor immunity and the omega 6:3 ratio with the score for promotion of tumor immunity (Fig. 5b, Bonferroni-corrected $P < 0.01$ for all).”

Reviewer 3

Authors determined the levels of 24 fatty acids in serum samples of prostate cancer patients (PCA) and healthy volunteers (control) from Ghana and United States (African American and European American were represented at similar number). This allowed the authors to stratify the patients into three ethnic categories: Ghanaian, African American and European American. The main finding is that American man show higher level of trans fatty acids, and the risk of developing prostate cancer is correlated with elevated trans fatty acid levels. In other words, they confirmed a commonly accepted information that trans fatty acid high levels correlate with the risk of prostate cancer.

They also found that in Ghana men, omega-3 fatty acids are present at higher levels than in American men, regardless their African or European ancestry. Omega-3 fatty acids are considered to show a potentially protective effect against prostate cancer [Cancer Epidemiol. Biomark. Prev. 2013, 22, 697–707].

Main concerns:

Comment: The title is not informative.

Response: The maximum length of a title in the journal is 15 words – as stated. Staying within the limit does not allow for details. To address the raised point, we revised the title to:

“Association of circulating fatty acids with socio-demographics, diet, FADS1/2 locus, and prostate cancer among Ghanaian, African-American, and European-American men”

Comment: *How the samples collection was planned is not explained in sufficient terms. The authors states: “Only population controls provided overnight-fasting blood”. Also taking the fasting/feeding status into account, comparing individual fatty acid levels in non-fasted prostate cancer patients with fasted healthy men (especially in fig 5. – volcano plot) seems risky*

Response: Thank you. In the NCI-Maryland study, cases and population controls had identical recruitment protocols. The participants were not asked to provide overnight-fasting blood. In the Ghana study, men without prostate cancer came for a routine check up to the hospital and were asked to provide overnight-fasting blood for glucose measurement and research

purposes, at time of recruitment. Ghanaian cases were not asked to provide overnight-fasting blood. We now acknowledge this limitation more prominently and revised the Methods section for clarity. Yet, we think our findings are valid. The pattern of the circulating fatty acid profiles in Ghanaian men were consistent with the reported profile of Nigerian men, and the association of *trans* fatty acids with prostate cancer was common to all three population groups, with a significant dose-dependent increase in the odds of having prostate cancer with increasing elaidic, palmitelaidic, and linoelaidic fatty acid concentrations in all three population groups.

Under Methods, we write:

“Serum sample processing. The participants in the two studies provided blood samples at time of recruitment. For the NCI-Maryland study, most blood samples were processed the same day, but always within 48 hours, after storage in a refrigerator. Both cases and controls provided non-fasting blood. For the NCI-Ghana study, blood samples were processed within 6 hours. In this study, controls were asked to provide overnight-fasting blood while cases were not asked to provide non-fasting blood. Serum was prepared using standard procedures and aliquots were stored at -80^o C. Serum samples were shipped from Ghana to the NCI in dry ice boxes.”

In the limitation section of the manuscript

“Moreover, the fact that only the controls for the Ghana study provided overnight-fasting blood is a limitation of our study.”

Comment: *Additionally, Fig2. Presents multiple factors that did (or did not) correlate with specific fatty acids with relation to age, BMI, education, aspirin use, smoking or diabetes, but NO info on how advanced the disease was. There is no information in the paper about stratification of patients according to i.e. Gleason score, at the moment of diagnosis/sample collection. Including this information would be a valuable addition to analysis, if the data are available.*

Response: Thank you. The reviewer raises questions similar to reviewer #1. Please see our response to reviewer #1, comment #2. We performed more analyses studying the relationship of circulating fatty acids with Gleason score, as suggested by both reviewers. However, these analyses did not reveal significant associations between them in our study population. In addition, the manuscript contains an analysis of the relationship between circulating fatty acids and NCCN risk score classification of men with prostate cancer. NCCN risk scores incorporate the Gleason score.

Comment: *Fig. 3 and Fig.4. are unclear to me. Fatty acid desaturases (encoded by the FADS1 and FADS2 genes) are enzymes involved in the synthesis of polyunsaturated fatty acids, namely in the conversion of linoleic acid into arachidonic acid (AA), and of α -linolenic acid (ALA) into eicosapentaenoic acid EPA and docosahexaenoic acid. Why is there in the fig 3 and 4 the whole*

list of fatty acids, including those that are not supposed to be metabolized by these enzymes, instead of the relevant substrates and products only?

Response: We agree with the reviewer but have a rationale of showing the data as is. Our data in Fig. 3 and 4 confirm that FADS1 & 2 only affect the levels of polyunsaturated fatty acids. We analyzed the effect of FADS1 & 2 on all the 24 fatty acids that we measured instead of just the polyunsaturated fatty acids to confirm if our observations are consistent with current knowledge. Indeed, FADS1 & 2 germline genetics only affected polyunsaturated fatty acids, as it should. We believe these data is important to show as is, to provide additional validity for our findings.

Reviewer 4

I would like to thank the authors for this important paper. This paper is important when considering the sample size, the population structure, and the quality of the analysis conducted.

MAJOR COMMENTS

Comment #1: *My main concern was with regards to the units of the fatty acids. Although I totally agree with the use of absolute vs. relative concentrations, it does not make much of a sense when concentrations are summed up. For example, what does the total of all fatty acids stand for? I would recommend that the authors keep the sub-total but remove the grand total of the fatty acids.*

Response: Thank you for this comment. We included “Total fatty acid” in our analysis because this value was provided to us by the service provider that measured the fatty acids (Omegaquant). We understand the concern, hence have removed the total fatty acid category throughout the manuscript, figures, and tables, supplementary tables, and supplementary figures.

Comment #2: *I suspected that the authors presented OR for prostate cancer by unit increment. This is not correct and is likely explaining results such as OR and 95%CI of 1.00 (1.00-1.00) $P < 0.001$ (table 2). The results in Table 2 should be rather presented for log-transformed values of the fatty acids, or (I would prefer) by standard deviation increment for each fatty acid.*

Response: We thank the reviewer for the comment. We believe the reviewer is referring to Table 1. Per recommendation, we revised Table 1 where now the ORs represent an increase in estimated odds of prostate cancer per one standard deviation increase for each fatty acid.

Comment #3: Regarding trans fatty, I would strongly recommend to the authors to further consider the subdivision, ruminant trans and industrial trans for two main reasons. We now know that both industrial and ruminant fatty acids have divergent dietary sources, and often dissimilar associations with diseases.

Response: We thank the reviewer for bringing this to our attention. Although these different types of trans fatty acids may have dissimilar associations with other diseases, in our analysis, we did not see divergent associations with prostate cancer risk or aggressiveness, hence we did

not find the need to create further subdivisions. However, to further address this point we write now as follows in the revised manuscript, in the discussion, on page 13.

“Circulating trans-fatty acids can have various sources. In an Asian Indian population, it was shown that heating and frying with edible oils (including refined soybean oil, groundnut oil, olive oil, and rapeseed oil) resulted in a significant increase in the amount of both trans and saturated fatty acids from these oils³⁵. In addition, a cross-sectional study in Uganda revealed high proportions of saturated, cis-monounsaturated, and trans fatty acids, likely due to the high consumption of palm oil and hydrogenated fats³⁶. In the United States and many other countries, there are two main types of trans fatty acids: industrial and natural (ruminant) trans fatty acids. Industrial trans fatty acids, such as elaidic acid, are found in processed foods that contain partially hydrogenated vegetable oils, which are used in frying oils, margarines, spreads, and in bakery products³⁷. Because the level of elaidic acid in the blood correlates with intake of highly processed foods, elaidic acid is considered a marker of processed food intake³⁸. On the other hand, natural (ruminant) trans fatty acids, such as palmitelaidic acid, are made by bacterial metabolism of polyunsaturated fatty acids in the rumen of ruminants such as cows, sheep, and goats, and are present in fats sourced from these animals³⁷. In our study, frequent intake of bacon fat or drippings partly explained the variability in the concentration of both types of trans fatty acids. However, we did not find that dietary intake had divergent associations with industrial vs. natural (ruminant) trans fatty acid. This is likely because our survey may not have adequately captured the intake of highly processed foods. Additionally, industrial and natural (ruminant) trans fatty acids have been reported to have dissimilar associations with some diseases such as coronary heart disease³⁹, diabetes^{40,41}, colorectal cancer⁴², and pancreatic cancer⁴³. However, in our study, both types of trans fatty acids showed similar associations with prostate cancer risk and aggressiveness.”

Comment #4: Overall, the discussion section needs to be improved. First, the authors did not comment on the inverse association observed with EPA, DPA, and DHA. This is equally an important finding and may allude to a possible beneficial effect of fish consumption? Second, the authors should comment separately on elaidic acid and palmitelaidic acid. Elaidic acid is a marker of processed food intake (PMID: 25675445) and is considered as an industrial trans whereas palmitelaidic is a ruminant trans fatty acid (PMID: 33913149).

Response: We only commented on the association of trans fatty acids with prostate cancer because those were the fatty acids that showed consistent associations with prostate cancer risks in all three population groups – and wanted to stay focused. We changed the discussion section as described in our response to comment #3. For EPA, DPA, and DHA, we added the following paragraph on pages 8-9 to the revised manuscript:

“Notably, three of the four omega-3 fatty acids, namely docosahexaenoic (DHA), docosapentaenoic-n3 (DPA), and eicosapentaenoic (EPA) acids, were inversely associated with prostate cancer among Ghanaian men (Table 1, Fig. 4a). On the other hand, several omega-6 fatty acids positively associated with prostate cancer among EA men (Table 1, Fig. 4c). However, only the three measured trans fatty acids, namely elaidic, palmitelaidic, and

linoelaidic acids, were positively associated with the odds of having prostate cancer in all men combined and across the three population groups (Fig. 4a-c)."

Comment #5: *The authors should provide the full nomenclature of the fatty acids, especially in Supp Table 2 and appropriately within the manuscript from line 386 (see PMID: 28465289)*

Response: We appreciate this comment. We now report the correct nomenclature for all the fatty acids in Supplementary Table 2 and in the method section under "Serum fatty acid measurement".

We write:

"The following 24 fatty acids (by class) were identified: saturated (14:0, 16:0, 18:0, 20:0, 22:0, 24:0); trans (16:1n-7t, 18:1n-9t, 18:2n-6t, 9t); cis monounsaturated (16:1n-7, 18:1n-9, 20:1n-9, 24:1n-9); cis n-6 polyunsaturated or omega-6 (18:2n-6, 18:3n-6, 20:2n-9, 20:3n-6, 20:4n-6, 22:4n-9, 22:5n-9); and cis n-3 polyunsaturated or omega-3 (18:3n-3ccc, 20:5n-3, 22:5n-3, 22:6n-3) (Supplementary Table 2)."

MINOR COMMENTS

Comment #6: *About some syntaxes*

- It is preferred to state that a risk factor is associated with high or low risk, not "increased" or "decreased" risk, which suggests a referential start for the risk.

Response: Thank you. We made the change at several places but not always. In this manuscript, the reference to risk commonly comes from an odds ratio, indicating an increased risk (or increased odds) with reference to a comparison group. Saying "high risk" indicates a judgement that not all readers may agree to.

Comment #7: *The authors routinely used "yet" and "still" especially at the beginning of sentences, although the link of these adverbs with preceding affirmations were unclear. Lines 51, 60, 163, 238, 253...should be re-arranged*

Response: This has been revised in the manuscript.

Comment #8: *The authors should avoid some generic sentences, which in my opinion were misleading. Line 284: "Fatty acid levels are under genetic control". This is not necessarily true. Many fatty acids are directly sourced from the diet, and their circulating levels confirm past intakes. Another generic sentence: "Our study has strengths and limitations". This is an empty sentence.*

Response: We have either revised or removed these sentences.

ABSTRACT

Comment #9: *The first sentence is utterly wrong. It assumed a positive association for a group of molecules with disparate structures and functions. The association between fatty acids and cancers is specific for each fatty acid and cancer site.*

Response: We thank the reviewer for this comment. We revised the abstract accordingly.

The revised abstract reads as follows:

“The association between fatty acids and prostate cancer remains poorly explored in African-descent populations. Here, we analyzed 24 circulating fatty acids in 2,934 men, including 1,431 prostate cancer cases and 1,503 population controls from Ghana and United States, using CLIA-certified mass spectrometry-based assays. We investigated their associations with population groups (Ghanaian, African-American, European-American men), lifestyle factors, the fatty acid desaturase (FADS) genetic locus, and prostate cancer. Blood levels of circulating fatty acids varied significantly between the three population groups, particularly trans, omega-3 and omega-6 fatty acids. FADS1/2 germline genetic variants and lifestyle factors explained some of the variation in fatty acid levels, with the FADS1/2 locus showing population-specific associations, suggesting differences in their control by germline genetic factors. All trans fatty acids, namely elaidic, palmitelaidic, and linoelaidic acids, were associated with increased odds of prostate cancer, independent of ancestry or geographic location, or potential confounders.”

Comment #10: Line 45: replace examined by "analysed", the sentence is referring to the laboratory work.

Response: This replacement has been made.

Comment #11: Line 51: please make that sentence clear and straightforward. Just state the associations between TFA and prostate cancer.

Response: We revised the abstract and shortened it to less than 150 words, to follow journal requirements and style. We made it clearer and straightforward. See response to comment #9 for revised abstract.

INTRODUCTION

Comment #12: Line 69-74: there is a need to improve that section. The authors rightly cited two meta-analyses that investigated the relationships between fatty acids and prostate cancer, one on dietary fatty acids and a second on circulating levels. Then, the authors went on to elaborate on a result of a case-cohort study. This looks like cherry-picking to me. The authors should rather comment on the meta-analyses, especially on circulating levels, and succinctly present known associations with prostate cancer.

Response: Thank you. On page 3, we revised the section as follows:

“The role of fatty acids in prostate cancer has been studied extensively, but the observations are conflicting, and a consensus of the effects of fatty acids on prostate cancer risk has yet to be achieved^{5,6}. A meta-analysis on prospective studies investigating the association of 14 circulating fatty acids, namely saturated and mono- and polyunsaturated fatty acids, with prostate cancer risk in 5,098 cases and 6,649 controls reported an inverse association between stearic acid, a saturated fatty acid, and the risk to develop prostate cancer. From another meta-analysis, focusing on dietary trans-fatty acid, the authors concluded that an elevated total

intake of trans-fatty acids may increase prostate and colorectal cancer risks⁶. Other studies, using epidemiological and experimental approaches, linked the intake and synthesis of saturated fatty acid to advanced or fatal prostate cancer^{7,8}. Together, these studies support the involvement of fatty acids in prostate tumorigenesis and progression. Nevertheless, population differences with reference to race/ethnic groups were not sufficiently explored in these large studies because of a lack of diversity in the assessed populations.”

Comment #13: Line 78: It is preferable to say aim rather than goal

Response: We have revised this sentence accordingly.

Comment #14: Line 78-88: This is confusing. Are these the objectives? It sounds like results to me

Response: Thank you for this comment. Greatly appreciated. We have now revised this paragraph to more clearly state our aims. On pages 3-4, we write as follows:

“Being aware of this knowledge gap, we decided to characterize the relationship between circulating fatty acids and prostate cancer in the ethnically diverse NCI-Maryland and NCI-Ghana Prostate Cancer Case-Control studies, with an over-representation of men of African descent. We aimed to find out if common associations exist or if there are distinct patterns among Ghanaian, AA, and EA men. In addition, we explored how circulating fatty acid levels may relate to demographic, lifestyle, and germline genetics, and to an immune-oncology marker profile.”

RESULTS

Comment #15: Line 103: I would not say "global", you mean substantial?

Response: We removed global.

TABLE 1

Comment #16: Provide the number of cases/controls on top of the table

Response: We thank the reviewer for the recommendation. Population totals (case/control) for each group have been added to the top of Table 1.

Comment #17: By age of study entry, you mean age at recruitment? These are case-control studies, we mostly talk about entry for cohorts. Please correct throughout the manuscript

Response: This has been corrected to age at recruitment throughout the manuscript, figures, and tables.

TABLE 2

Comment #18: How are the tertiles created? Normally in case-control studies, the quantiles are created based on the controls, then transferred to the cases.

Response: The cutoff for the tertiles was determined using the distribution of fatty acids in the control population for each study (NCI-MD or NCI-Ghana). We have now added a clarifying sentence to the footnote of Table 2.

Comment #19: *It is not common practice to provide % of cases in such tables.*

Response: We agree with the comment. Thus, we have removed the % cases/controls in Table 2, as well as Supplementary Table 13.

Comment #20: *No need to provide P value, only P for trend is important. And it should be stated "P trend", not just trend.*

Response: We removed the P value column in Table 2 and replaced "Trend" with "P Trend"

Comment #21: *Median values of the concentration for each tertile should be provided*

Response: We thank the reviewer for this comment. We have added the median values to the revised Table 2.

METHODS

Comment #22: *Line 396: why "completely" randomised, please correct that*

Response: It has been corrected in the text.

SUPPLEMENTARY TABLE 1

Comment #23: *Why AA and EA below diabetes? Maybe this is just an issue with the preparation of the pdf?*

Response: Thank you for catching this error. This has been corrected.

DISCUSSION

Comment #24: *Line 254: Please check the paper in Uganda (PMID: 30477593) and in Asian Indians (PMID: 27374582) to discuss the sources of trans fatty acids in other populations*

Response: We thank the reviewer for the addition of these valuable studies. We have incorporated both of them into our discussion, on page 13, on sources for *trans* fatty acids.

We write as follows:

"Circulating trans-fatty acids can have various sources. In an Asian Indian population, it was shown that heating and frying with edible oils (including refined soybean oil, groundnut oil, olive oil, and rapeseed oil) resulted in a significant increase in the amount of both trans and saturated fatty acids from these oils³⁵. In addition, a cross-sectional study in Uganda revealed high proportions of saturated, cis-monounsaturated, and trans fatty acids, likely due to the high consumption of palm oil and hydrogenated fats³⁶...."

REVIEWERS' COMMENTS

Reviewer #1 (Remarks to the Author):

The authors have addressed my comments.

Reviewer #2 (Remarks to the Author):

Seeing the revisions and responses, it is still hard for me to justify the significance of the work as it is presented. The association between the trans fatty acids and prostate cancer is the key observation, which, however, is commonly accepted knowledge. Some other suggested revisions would require some significant work and could not be done.

Reviewer #4 (Remarks to the Author):

After careful examination of the submitted documents, I can assure that all my concerns have been addressed by the authors.

I do not have any additional comments.

NCOMMS-22-33513A

Point-by-point response to reviewers' comments

REVIEWERS' COMMENTS

Reviewer #1 (Remarks to the Author):

The authors have addressed my comments.

Response: We thank this reviewer for their time and comments to improve our manuscript.

Reviewer #2 (Remarks to the Author):

Seeing the revisions and responses, it is still hard for me to justify the significance of the work as it is presented. The association between the trans fatty acids and prostate cancer is the key observation, which, however, is commonly accepted knowledge. Some other suggested revisions would require some significant work and could not be done.

Response: We appreciate this reviewer's time and input on our manuscript. We believe that our manuscript describes the fatty acid profile in a large and unique study population of African and African American men, not only characterizing their association with prostate cancer, but also their relationship with immune-oncological markers and how that differs by prostate case status and population. Our study not only confirms previous knowledge of the association between prostate cancer and fatty acids, but also how this impacts the immune system in a diverse population.

Reviewer #4 (Remarks to the Author):

After careful examination of the submitted documents, I can assure that all my concerns have been addressed by the authors.

I do not have any additional comments,

Response: We would like to thank this reviewer for their time and commitment to the peer review process.